# Translational pharmacology of an inhaled small molecule αvβ6 integrin inhibitor for idiopathic pulmonary fibrosis

Alison E. John [1], Rebecca H. Graves [2], K. Tao Pun [2], Giovanni Vitulli [2], Ellen J. Forty[3], Paul F. Mercer[3], Josie L. Morrell[2], John W. Barrett[2], Rebecca F. Rogers [2], Maryam Hafeji [2], Lloyd I. Bibby[2], Elaine Gower[2], Valerie S. Morrison[2], Yim Man[2], James A. Roper [2], Jeni C. Luckett[4], Lee A. Borthwick[5], Ben S. Barksby[5], Rachel A. Burgoyne[5], Rory Barnes[5], Joelle Le[6], David J. Flint[7], Susan Pyne [7], Anthony Habgood [1], Louise A. Organ[1], Chitra Joseph [1], Rochelle C. Edwards-Pritchard [1], Toby M. Maher [8,9], Andrew J. Fisher[5,10], Natasja Stæhr Gudmann[11], Diana J. Leeming[11], Rachel C. Chambers [3], Pauline T. Lukey[2], Richard P. Marshall[2], Simon J. F. Macdonald[2], R. Gisli Jenkins [1✉] & Robert J. Slack [2]

The αvβ6 integrin plays a key role in the activation of transforming growth factor-β (TGFβ), a pro-fibrotic mediator that is pivotal to the development of idiopathic pulmonary fibrosis (IPF). We identified a selective small molecule αvβ6 RGD-mimetic, GSK3008348, and profiled it in a range of disease relevant pre-clinical systems. To understand the relationship between target engagement and inhibition of fibrosis, we measured pharmacodynamic and disease-related end points. Here, we report, GSK3008348 binds to αvβ6 with high affinity in human IPF lung and reduces downstream pro-fibrotic TGFβ signaling to normal levels. In human lung epithelial cells, GSK3008348 induces rapid internalization and lysosomal degradation of the αvβ6 integrin. In the murine bleomycin-induced lung fibrosis model, GSK3008348 engages αvβ6, induces prolonged inhibition of TGFβ signaling and reduces lung collagen deposition and serum C3M, a marker of IPF disease progression. These studies highlight the potential of inhaled GSK3008348 as an anti-fibrotic therapy.

[1] Respiratory Medicine NIHR Biomedical Research Centre, University of Nottingham, Nottingham, UK. [2] Fibrosis DPU, Respiratory TAU, GlaxoSmithKline, Stevenage, Hertfordshire, UK. [3] Centre for Inflammation and Tissue Repair, University College London, London, UK. [4] Radiological Sciences, University of Nottingham, Nottingham, UK. [5] Fibrosis Research Group, Newcastle University Biosciences Institute and Newcastle University Translational and Clinical Research Institute, Newcastle upon Tyne, UK. [6] Drug Design and Selection - Molecular Design, Respiratory TAU, GlaxoSmithKline, Stevenage, Hertfordshire, UK. [7] Strathclyde Institute of Pharmacy and Biomedical Sciences, University of Strathclyde, Glasgow, UK. [8] NIHR Respiratory Clinical Research Facility, Royal Brompton Hospital, London, UK. [9] Fibrosis Research Group, National Heart and Lung Institute, Imperial College, London, UK. [10] Institute of Transplantation, Freeman Hospital, Newcastle Upon Tyne Hospitals NHS, Foundation Trust, Newcastle upon Tyne, UK. [11] Nordic Bioscience A/S, Biomarkers and Research, Herlev Hovedgade 205-207, Herlev, Denmark. ✉email: gisli.jenkins@nottingham.ac.uk

Fibrosis is the formation of scar tissue and occurs due to injury or long-term inflammation followed by abnormal wound healing. It is a leading cause of morbidity and mortality in a range of diseases[1]. Fibrotic disorders include idiopathic pulmonary fibrosis (IPF), the most common of the idiopathic interstitial pneumonias, for which there are currently limited pharmacological therapeutic options[2]. The global incidence of IPF is increasing and is conservatively estimated to be 3–9 people per 100,000 per year[3]. Recently, there has been international approval of pirfenidone and nintedanib, drugs that slow disease progression in IPF[4–7]. However, the failure of pirfenidone and nintedanib to halt or reverse disease progression, and their unclear mechanisms of action make them sub-optimal treatments. Furthermore, their side-effect profiles lead to considerable patient tolerability issues[7,8]. Existing therapies are oral and one of the areas yet to be fully explored in IPF is the potential for the inhaled route of delivery directly to the fibrotic lung. By minimizing the drug dose and systemic exposure, topical delivery to the lungs could offer an improved safety profile compared with systemically bioavailable treatments.

The alpha-v beta-6 (αvβ6) integrin is a member of the arginyl-glycinyl-aspartic acid (RGD) sub-family of the heterodimeric, transmembrane glycoprotein receptors[9]. The primary function of the αvβ6 integrin is to activate the key pro-fibrotic mediator, transforming growth factor-β1 (TGFβ1)[10,11]. The αvβ6 integrin is upregulated in patients with IPF and the level of expression appears to be linked to prognosis[12,13], making this integrin not only a potential biomarker of disease progression, but also an attractive therapeutic target. The interest in αv integrins as therapeutic targets in fibrosis has increased significantly[14], with many pharmaceutical companies beginning multiple drug discovery initiatives within this space. This has been led by the development of the αvβ6 monoclonal antibody STX100 (known pre-clinically as 3G9 and clinically as BG00011)[12] that has recently completed a phase II trial for IPF[15]. The extensive target validation of αvβ6 within IPF, other types of organ fibrosis[16,17] and cancer[18], combined with the significant renewed interest in

the RGD integrins by academic and industrial groups[14,19], has resulted in a renaissance for integrins as drug targets.

To improve the probability of success and enable robust testing of early pre-clinical efficacy, a bespoke, IPF-specific, inhaled drug discovery effort targeting the αvβ6 integrin was carried out. A clear path from pre-clinical to early clinical studies was designed to ensure target engagement could be measured and associated with functional, innovative pharmacodynamic (PD), and disease end points, including imaging and soluble biomarkers to increase the chance of successful translation into clinical trials. In this study, the pre-clinical characterization of an inhaled first in class clinical candidate, GSK3008348, is described. These studies demonstrate that GSK3008348 binds to the αvβ6 integrin with high affinity and selectivity in human fibrotic lung tissue and isolated lung epithelial cells. GSK3008348 inhibits TGFβ activation and reduces downstream pro-fibrotic signaling. In the murine bleomycin-induced model of lung fibrosis, GSK3008348 engages αvβ6 integrins, inhibits the activation of TGFβ with a prolonged duration of action, and reduces lung fibrotic end points including collagen deposition and serum C3M levels, a clinically relevant marker of IPF disease progression. These studies describe an exemplar pathway for the development of an inhaled αvβ6 integrin inhibitor, highlighting the potential of inhaled GSK3008348 as an anti-fibrotic therapy.

## Results

**Physico-chemical and pharmacokinetic profile of GSK3008348.** GSK3008348 is an RGD-mimetic with the tetrahydronaphthyridine and arylbutanoic acid motifs replacing the arginine and aspartic acid residues, respectively (Fig. 1a). Molecular modeling was used to aid the structure activity relationships around GSK3008348 based on a homology model developed using the closed form of αvβ3 X-ray structure 1L5G[20] as a template. Docked GSK3008348 (orange) predicts binding site electrostatic interactions between the 1,8-tetrahydronaphthyridine and αv-Asp218, coordination of the carboxylic acid motif with a manganese ion in the metal ion dependent

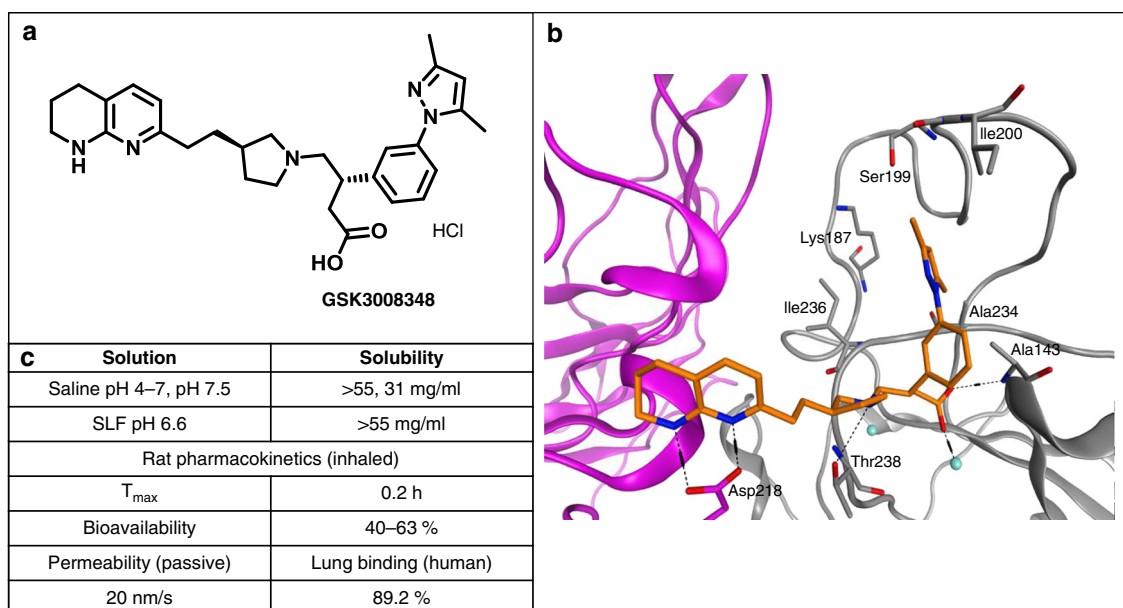

**Fig. 1 Structure, solubility, and pharmacokinetic properties of GSK3008348. a** The chemical structure of GSK3008348 with full chemical name: (*S*)-3-(3-(3,5-dimethyl-1*H*-pyrazol-1-yl)phenyl)-4-(*R*)-3-(2-(5,6,7,8-tetrahydro-1,8-naphthyridin-2-yl)ethyl)pyrrolidin-1-yl)butanoic acid. **b** Docked GSK3008348 (orange) in a homology model of αvβ6 based on αvβ3 X-ray (1L5G) composed of α-subunit (magenta) and β-subunit (gray). **c** Solubility and key pharmacokinetic properties for GSK3008348 are also shown. SLF, simulated lung fluid.

**c**

| Solution | Solubility |
| --- | --- |
| Saline pH 4–7, pH 7.5 | >55, 31 mg/ml |
| SLF pH 6.6 | >55 mg/ml |
| Rat pharmacokinetics (inhaled) | |
| $T_{max}$ | 0.2 h |
| Bioavailability | 40–63 % |
| Permeability (passive) | Lung binding (human) |
| 20 nm/s | 89.2 % |

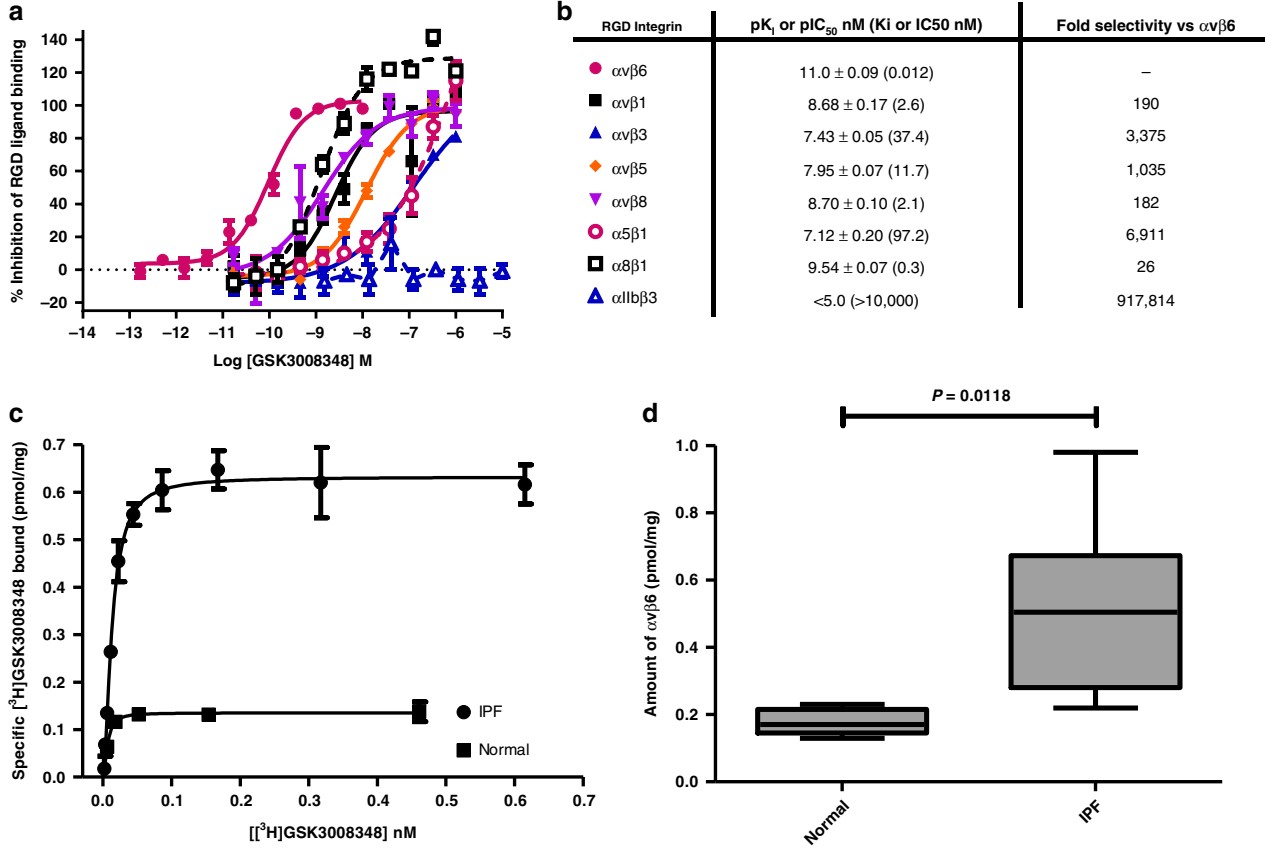

**Fig. 2 High affinity and selectivity of GSK3008348 for the αvβ6 integrin. a** Competition binding curves of GSK3008348 against the RGD integrins (mean ± SEM; $n = 4$). **b** $pK_I$ affinity or $pIC_{50}$ values (αIIbβ3 only) for GSK3008348 against the RGD integrins with fold-selectivity for αvβ6 (mean ± SEM; $n = 4$-6). **c** Representative saturation binding curves for $[^3H]GSK3008348$ in normal and IPF human lung tissue (duplicate points shown ± min/max). **d** Quantification of total levels of αvβ6 levels in normal and IPF human lung tissue (Box and whisker plot shows median, first and third quartiles, and max/min values; six donors per group). Source data are provided as a Source Data file.

adhesion site and an H-bond with Ala143 (Fig. 1b). The dimethyl-pyrazolo phenyl moiety binds in proximity to β6 subunit residues Ala143, Lys187, Ser199, Ile200, Ala234, and Ile236 in the specificity binding loop (SDL) region.

Crystalline GSK3008348 hydrochloride was prepared in seven linear steps from commercially available starting materials with the key synthetic step featuring a stereoselective introduction of the pyrazoloaryl ring via a rhodium catalyzed boronic acid addition to a crotonate[21]. The physico-chemical properties are commensurate with inhaled dosing with a measured moderate lipophilicity of chrom logD 2.77 and high solubility (>10 mg/ml) reflecting the presence of ionizable functionality. Pharmacokinetic (PK) studies show GSK3008348 has moderate permeability, a favorable distribution to the site of action, high clearance and low oral bioavailability, all appropriate for inhaled dosing (Fig. 1c) (see supplementary methods section for PK methods).

**αvβ6 affinity and RGD integrin selectivity of GSK3008348.** In recombinant soluble protein preparations, GSK3008348 exhibits a high affinity and a minimum of 26-fold selectivity for the αvβ6 integrin over the other RGD integrin family members (Fig. 2a and b). For the TGFβ-activating RGD integrins (αvβ1, αvβ3, αvβ5, and αvβ8), a minimum selectivity of 182-fold (αvβ8) and maximum selectivity of 3375-fold (αvβ3) was demonstrated (Fig. 2b). Of particular note, the inhibition constant for binding to the αvβ6 integrin was 190-fold higher than for the αvβ1 integrin and over a 1000-fold higher than for the αvβ3 or αvβ5 integrins (Fig. 2b). Plasma membranes prepared from normal and IPF human lung

tissues were used to investigate the affinity of $[^3H]GSK3008348$ for αvβ6 integrins in a disease relevant system as well as to quantify differences in the αvβ6 integrin expression between normal and diseased lung. $[^3H]GSK3008348$ demonstrated a high affinity for the αvβ6 in normal ($pK_D$ 11.0 ± 0.06 ($K_D$ 9.5 pM)) and IPF ($pK_D$ 11.1 ± 0.07 ($K_D$ 4.8 pM)) human lung tissue membrane preparations (Fig. 2c). A significant increase in the amount of αvβ6 integrin was observed in IPF membranes (0.51 ± 0.11 pmol/mg) compared with normal membranes (0.18 ± 0.02 pmol/mg) (Fig. 2d).

**GSK3008348-induced αvβ6 internalization and degradation.** The surface (membrane) and total (membrane and intracellular pools) expression of αvβ6 integrins in normal human bronchial epithelial (NHBE) cells were measured using flow cytometric and imaging assays. GSK3008348 caused a significant reduction in surface expression of αvβ6 (Fig. 3a). When cells were permeabilized with saponin to allow antibodies intracellular access, there was no significant difference observed between anti-hβ6 antibody labeling with and without GSK3008348 (Fig. 3a). The internalization observed in the presence of GSK3008348 was inhibited when NHBE cells were pre-incubated with the clathrin-coated pit inhibitor chlorpromazine, whereas the lipid raft inhibitor filipin had no effect showing endocytosis was mediated via clathrin-coated pits (Fig. 3b). To determine the rates of ligand-induced internalization the surface expression of αvβ6 integrin was measured over time following the addition of a maximal concentration of GSK3008348. GSK3008348 caused a rapid internalization

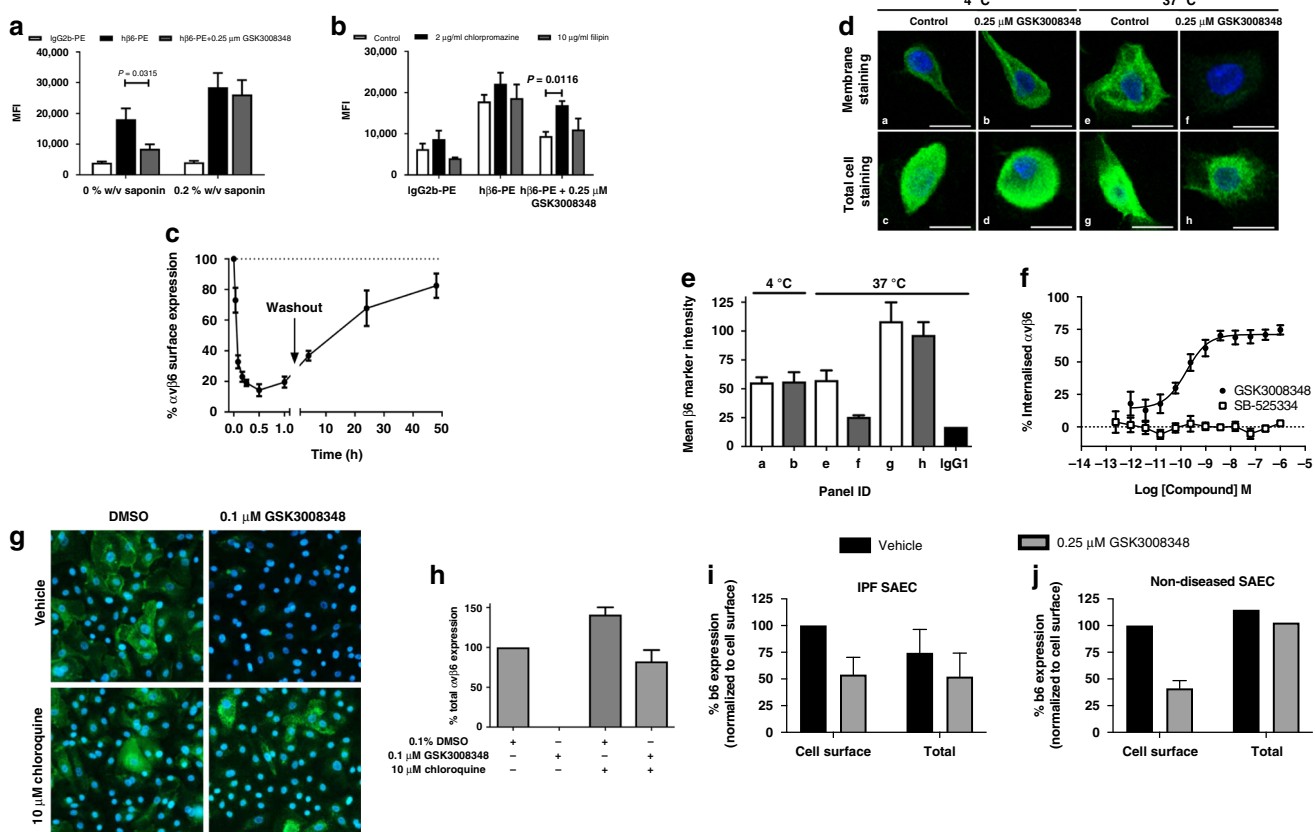

**Fig. 3 GSK3008348-induced internalization and degradation of the αvβ6 integrin.** Flow cytometry using NHBE cells showing **a** ligand-induced αvβ6 internalization triggered by GSK3008348, **b** the effect of chlorpromazine and filipin on ligand-induced αvβ6 internalization (mean ± SEM; $n = 4$–6; ANOVA with Tukey's post test comparisons versus hβ6-PE or control), and **c** the kinetics of GSK3008348-induced αvβ6 internalization and return post-washout determined via flow cytometry in (mean ± SEM; $n = 4$). **d** Localization of αvβ6 integrin in NHBE cells post-GSK3008348 addition (β6 integrin (green) and nucleus (blue)) imaged by confocal microscopy to produce a 3D cell z-stack. Representative single cells (×50 magnification; scale bar = 50 μm) from four optical sections captured per condition with **e** the mean β6 marker intensity for optical sections shown (mean ± SEM from single experiment). **f** Flow cytometric analysis of the concentration-dependency of GSK3008348-induced αvβ6 internalization (mean ± SEM; $n = 4$). **g** The effect of the lysosomal degradation inhibitor chloroquine on GSK3008348-induced αvβ6 degradation in NHBE cells measured by high-content screening with example images and **h** quantification of αvβ6 via image analysis (×10 objective with average of five optical fields) (mean ± SEM; $n = 5$). Ligand-induced αvβ6 internalization in SAECs from **i** IPF ($n = 2$) and **j** non-diseased patients ($n = 2$) at 1 h. Source data are provided as a Source Data file.

of αvβ6 integrins with a $t_{1/2}$ of 2.6 ± 0.5 min (Fig. 3c). Following washout of GSK3008348 after 1 h exposure, αvβ6 integrin returned to the surface of NHBE cells in a time-dependent manner with a $t_{1/2}$ of 11.0 ± 1.9 h (Fig. 3c).

The internalization of cell surface αvβ6 integrin was confirmed using confocal microscopy by immunofluorescence staining of the β6 subunit (Fig. 3d). Blocking internalization at 4 °C revealed abundant cell surface αvβ6 integrin expression (Fig. 3d panels a and b) and intracellular staining in cells permeabilized prior to staining with anti-hβ6 antibody (Fig. 3d panels c and d). Staining patterns were similar irrespective of the presence of GSK3008348. In contrast, although vehicle controls treated cells incubated at 37 °C retained cell surface expression of αvβ6 integrin (Fig. 3d panel e), those incubated with GSK3008348 for 1 h expressed no αvβ6 integrin on the cell surface (Fig. 3d, f) and all the intracellular integrin showed juxtanuclear localization (Fig. 3d panel h). To quantify the loss of αvβ6 integrin membrane staining induced by GSK3008348 in confocal microscopy studies, the mean staining intensity in chamber slide wells for anti-hβ6 antibody in the four optical sections captured were calculated (Fig. 3e). In flow cytometric assays, GSK3008348 caused concentration-dependent ligand-induced αvβ6 integrin internalization following 2 h incubation with NHBE cells with an $pEC_{50}$ value of 9.76 ± 0.25

($EC_{50}$ 0.26 nM) (Fig. 3f). No αvβ6 integrin internalization was observed with the TGFβR1 inhibitor SB-525334 (Fig. 3f). Further imaging studies using high-content screening were completed to measure total αvβ6 integrin levels in the absence and presence of the lysosomal degradation inhibitor chloroquine. Following a 24 h incubation with GSK3008348, the levels of staining for αvβ6 integrin was significantly reduced compared with control (Fig. 3g, h). The effect of GSK3008348 was reduced in the presence 10 μM chloroquine, demonstrating that GSK3008348-induced lysosomal degradation of αvβ6 post internalization. A trend toward increased levels of αvβ6 integrin in vehicle control treated cells exposed to the lysosomal inhibitor was also observed suggesting inhibition of endogenous turnover of the integrin in this system (Fig. 3g, h).

To understand whether αvβ6 integrin internalization may be disrupted in pulmonary fibrosis, limiting efficacy GSK3008348, internalization studies were performed in small airway epithelial cells (SAECs) from patients with IPF (Fig. 3i) or non-diseased controls (Fig. 3j). In both diseased and non-diseased SAECs there was loss of cell surface αvβ6 integrin 1 h following ligation with GSK3008348 to ~40–50% baseline levels. Whilst there was little difference in αvβ6 integrin internalization between diseased and non-diseased SAECs, there was a trend toward reduced total αvβ6

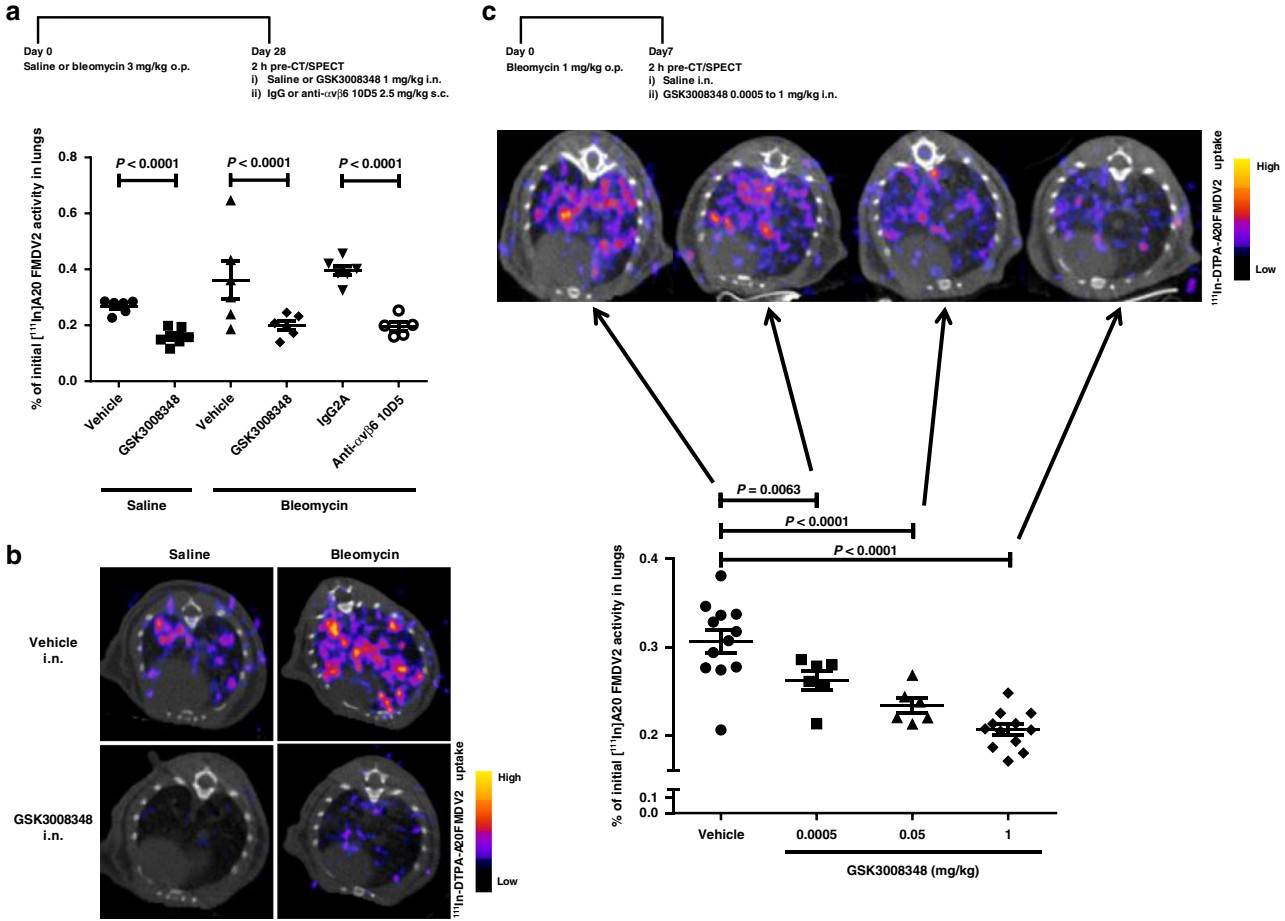

**Fig. 4 In vivo engagement of αvβ6 via the inhaled route in a fibrotic lung. a** Effect of GSK3008348 on the binding of the selective αvβ6 integrin nanoSPECT-CT ligand [$^{111}$In]A20FMDV2 in naïve (saline treated) and bleomycin-treated mice following i.n. dosing (mean ± SEM; n = 5–6 animals per group; ANOVA with Fisher's LSD post test comparisons versus corresponding vehicle or IgG2A control or between active groups). **b** Representative axial SPECT/CT scans of thorax detected binding of [$^{111}$In]A20FMDV2 in lungs of naïve (saline treated) or bleomycin-treated mice receiving vehicle (i.n. saline) or GSK3008348 (1 mg/kg) from the study shown in panel **a**. **c** Dose-dependent effect of GSK3008348 on the binding of the selective αvβ6 integrin nanoSPECT-CT ligand [$^{111}$In]A20FMDV2 in bleomycin-treated mice following i.n. dosing (mean ± SEM; n = 6–12 animals per group; ANOVA with Fisher's LSD post test comparisons versus vehicle i.n. control). Representative axial SPECT/CT scans of thorax detected binding of [$^{111}$In]A20FMDV2 also shown. Source data are provided as a Source Data file.

integrin in IPF derived SAECs that may reflect the different metabolic profile seen in these cells.

**NanoSPECT-CT imaging of GSK3008348 engagement with αvβ6.** We have previously shown upregulation of the αvβ6 integrin in bleomycin-induced pulmonary fibrosis using a non-invasive imaging (single photon electron computed tomography (SPECT)) of a highly selective radioligand for the αvβ6 integrin ([$^{111}$In]-DTPA-A20FMDV2)[22]. Therefore, this system was used to measure the engagement of GSK3008348 with αvβ6 integrins in the lung in vivo. GSK3008348 caused complete inhibition of αvβ6 integrin specific radioligand binding in both saline control and bleomycin-treated mice 2 h following dosing (Fig. 4a, b). This confirmed that GSK3008348 could completely inhibit radioligand binding via inhaled delivery, even when αvβ6 integrins have been upregulated in fibrotic mouse lung. Dose-dependent inhibition of radioligand binding 2 h post-intranasal (i.n.) dosing was also observed with 0.05 μg/kg and 50 μg/kg of GSK3008348 (Fig. 4c). In an additional study in naïve mice a significant inhibition of radioligand binding was also observed 8 h post-i.n. dosing of 1 mg/kg GSK3008348 (Supplementary Fig. 1). In PK studies completed in C57BL/6 mice dosed with 1 mg/kg i.n. GSK3008348, lung levels of drug were shown to be below the lower limit of

quantitation (2 ng/lung (~4 nM)) from 4 h post dosing (Supplementary Table 1).

**Inhibition of αvβ6-mediated TGFβ activation by GSK3008348.** To demonstrate the functional activity of GSK3008348, NHBE cells were used to measure TGFβ activation. GSK3008348 inhibited Smad2 phosphorylation in a concentration-dependent manner, as did the TGFβR1 inhibitor SB-525334 and the αvβ6 integrin selective peptide A20FMDV2 (Fig. 5a). The maximal level of inhibition achieved for GSK3008348, SB-525334, and A20FMDV2 were comparable showing that TGFβ activation in this system was mediated via αvβ6 integrins. The pIC$_{50}$ (IC$_{50}$) values for inhibition of αvβ6-mediated TGFβ activation were 9.13 ± 0.04 (0.15 nM), 8.25 ± 0.14 (3.5 nM), and 7.28 ± 0.13 (26.8 nM) for GSK3008348, A20FMDV2, and SB-525334, respectively. To further investigate the mechanism by which GSK3008348 caused a prolonged inhibition of αvβ6-mediated TGFβ release from NHBE cells, washout studies were completed in the presence and absence of the lysosomal degradation inhibitor chloroquine. GSK3008348 caused inhibition of TGFβ activation in the presence and absence of chloroquine (Fig. 5b). In the absence of chloroquine, the inhibitory effect observed for GSK3008348 was only partially reversed following washout. The partial reversal of

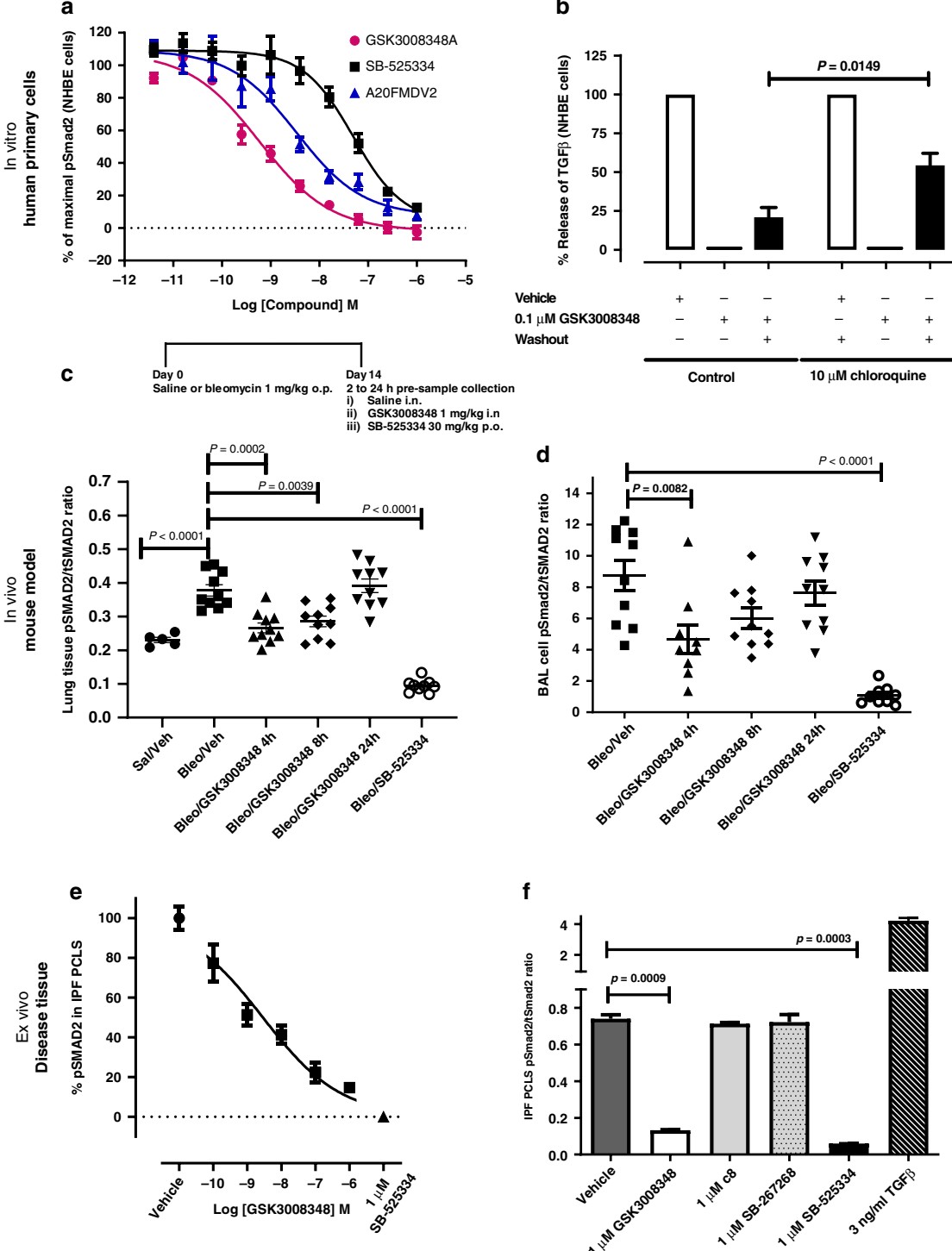

**Fig. 5 Inhibition of αvβ6-mediated TGFβ activation in vitro and in vivo by GSK3008348. a** Concentration-dependent inhibition of pSmad2 levels in NHBE cells by GSK3008348, SB-525334 (TGFβR1 inhibitor), and A20FMDV2 (selective αvβ6 peptide) (mean ± SEM; $n = 4$). **b** Duration of action of the inhibition of GSK3008348 on TGFβ-activation, and the effect of the lysosomal degradation inhibitor chloroquine, measured in a NHBE cell and TMLC (expressing the firefly luciferase under the control of a TGFβ-sensitive portion of the PAI-1 promoter) co-culture system (mean ± SEM; $n = 4$–8; Student's $t$ test). Effect of GSK3008348 (i.n.) versus SB-525334 (p.o.) on the levels of pSmad2 in **c** lung tissue and **d** BAL cells from bleomycin-treated mice (mean ± SEM; $n = 5$–10 animals per group; ANOVA with Fisher's LSD post test comparisons). **e** Concentration-dependent effect of GSK3008348 on pSmad2 levels in IPF PCLSs (mean ± SD; three PCLSs in a single donor representative of four donors). **f** Levels of pSmad2 in PCLS from an individual IPF donor in the presence of αv integrin tool inhibitors (mean ± SEM; three PCLSs in a single donor representative of four donors). *Sal/Veh* Saline/Vehicle, *Bleo/Veh* Bleomycin/Vehicle. Source data are provided as a Source Data file.

GSK3008348 inhibition caused by washout in the absence of chloroquine was significantly reduced in the presence of chloroquine (Fig. 5b).

To measure the functional inhibition of αvβ6-mediated TGFβ over time in vivo, pSmad2 levels were measured in both lung tissue and cells present in bronchoalveolar lavage (BAL) following administration of GSK3008348. Mice challenged with 1 mg/kg (20IU) bleomycin showed an increase in the pSmad2 levels in the lungs after 14 days relative to saline controls. (Fig. 5c). Following a single therapeutic i.n. dose of GSK3008348 (1 mg/kg) in bleomycin-challenged animals, a significant reduction in pSmad2 was observed in lung tissue when compared with saline treated animals at 4 and 8 h post dosing (Fig. 5c) despite levels of drug being unmeasurable in the lung 2 h post dosing (Supplementary Table 1). By 24 h the pSmad2 in the bleomycin/GSK3008348-treated animals had returned to levels comparable with those observed in the lung tissue from bleomycin/vehicle-treated animals (Fig. 5c). The percentage inhibitions at 4, 8, and 24 h were calculated to be 75%, 62%, and 9%, respectively. These findings were also observed in the BAL cells of these same animals, where GSK3008348 caused a reduction in pSmad2 in BAL cells from bleomycin-treated animals compared with BAL cells from bleomycin/vehicle-treated control animals. By 24 h, the levels of pSmad2 in the BAL cells from bleomycin/GSK3008348-treated animals had returned to levels comparable with those observed in BAL cells from bleomycin/vehicle-treated animals (Fig. 5d). The TGFβR1 inhibitor, SB-525334 was shown to significantly reduce pSmad2 levels in both lung tissue and BAL cells 2 h post dosing (Fig. 5c, d). Levels of lung tissue pSmad2 in the bleomycin/TGFβR1 inhibitor group were significantly lower than in the saline/vehicle group, suggesting that under normal conditions there is a basal level of TGFβ activity in the lungs. However, GSK3008348, which reduces active TGFβ levels through αvβ6 inhibition, did not inhibit pSmad2 below this basal activity at the dose tested. The pattern of dose-dependent inhibition of pSmad2 in bleomycin-challenged animals with GSK3008348 was comparable to that observed in CT/SPECT studies (Fig. 4 and Supplementary Fig. 2).

To provide a link between pre-clinical and clinical studies, the effect of GSK3008348 was measured in precision cut lung slices (PCLS) from IPF patients. GSK3008348 caused a concentration-dependent reduction in pSmad2 phosphorylation (Fig. 5e) with an approximate $IC_{50}$ of 3 nM. Levels of lung tissue pSmad2 following inhibition of TGFβR1 with SB-525334 in murine and human fibrosis was lower than inhibition by GSK3008348, suggesting that under normal conditions there is a basal level of TGFβ activity in the lungs which was not inhibited by GSK3008348 at the concentrations tested. It is possible that some of the residual TGFβ activity in the lung is due to activation by the αvβ1 integrin, therefore the αvβ1 inhibitor c8, and the αvβ3/αvβ5 inhibitor SB-267268, were assessed. Neither compound inhibited pSmad2 levels (Fig. 6f). These data suggest that in this system, the αvβ6 integrin is both necessary and sufficient for activating TGFβ in fibrotic lung.

**GSK3008348 effects on PD and disease biomarkers**. To further clarify the drug effects on PD and disease biomarkers, a number of analytes that are known to reflect fibrogenic end points, including analytes previously identified in the PROFILE (Prospective Observation of Fibrosis in the Lung Clinical End points) study[23] were investigated in a murine pulmonary fibrosis model. Animals were exposed to 3 mg/kg (60IU) bleomycin for 14 days following subcutaneous (s.c.) implantation of osmotic pumps containing GSK3008348 or vehicle 3 days before. Bleomycin treatment increased both total lung collagen as measured by

hydroxyproline levels (Fig. 6b) and serum levels of the matrix metalloprotease (MMP)-degraded ECM protein neoepitope C3M (Fig. 6c), which is known to reflect progressive IPF in patients. There was partial inhibition of hydroxyproline levels in the lungs of bleomycin-treated mice following prophylactic administration of GSK3008348 (Fig. 6b) but a substantial inhibition of serum C3M was detected in response to GSK3008348 (Fig. 6c).

TGFβ activation was also assessed in the lungs of IPF patients following collection of bronchoalveolar lavage fluid (BALF). Samples were obtained from healthy volunteers and IPF patients and the level of pSmad2 in BAL cells was measured along with TGFβ-induced gene h3 (βIG-H3) and plasminogen activator inhibitor-1 (PAI-1) within the BALF. All three biomarkers were significantly increased in IPF patients compared with healthy volunteers (Fig. 6d, e). To determine whether these markers could serve as PD biomarkers in a clinical study using GSK3008348 we also measured these analytes in samples collected from the study assessing the effects of prophylactic dosing with GSK3008348 in the bleomycin model of pulmonary fibrosis. There was a significant increase in lung tissue pSmad2 levels 14 days post instillation of bleomycin and phosphorylation of Smad2 was inhibited by GSK3008348 (Fig. 6f). Furthermore, GSK3008348 also inhibited bleomycin-induced increases in the level of βIG-H3 and PAI-1 detected in BALF (Fig. 6g). Finally, to determine whether local delivery of inhaled αvβ6 inhibitor could be used to ameliorate established fibrotic lung disease, therapeutic dosing of i.n. GSK3008348 was undertaken. Bleomycin-induced pulmonary fibrosis was induced with a 3 mg/kg oropharyngeal dose and after 14 days, at the time of established fibrosis, GSK3008348 was instilled twice daily i.n. for a further 14 days and fibrosis was assessed (Fig. 7a). Animals receiving vehicle control showed an increase in hydroxyproline (Fig. 7b) and areas of focal fibrosis assessed by histology (Fig. 7c) where those receiving twice daily i.n. αvβ6 inhibitor showed a significant reduction in fibrosis as measured by both hydroxyproline and histology (Fig. 7b, c).

### Discussion

The small molecule RGD-mimetic αvβ6 inhibitor GSK3008348 represents a therapeutic agent for inhaled delivery to IPF patients and is the first in class inhaled integrin inhibitor to reach clinical development. GSK3008348 resides in a distinct niche, not only in the current IPF therapeutic pipeline space, but also as one of the highest affinity integrin inhibitors that has been developed in the field of small and large molecule drug discovery[14]. The aim of this study was to fully characterize GSK3008348 and generate translational drug discovery biomarkers that will facilitate the performance of future clinical trials. To date, GSK3008348 has been shown to be safe and well tolerated in phase I healthy volunteer studies[24] and has completed a phase Ib study, demonstrating target engagement of the αvβ6 integrin in IPF patients[25].

It is hypothesized that the αvβ6 integrin is a key driver of TGFβ activation in IPF and therefore an attractive therapeutic target in this, and other, fibrotic diseases[10–13]. A highly selective αvβ6 integrin inhibitor was designed for topical delivery to the lungs to improve target engagement, reduce the dose, and minimize systemic exposure and potential adverse effects. A concern with this approach has been the risk of being unable to deliver drug to the peripheral regions of the lung where the fibrosis occurs in IPF, however, a recent study using [99m]Technetium-labeled salbutamol has shown that nebulized drug can be delivered to these fibrotic regions[26]. During development of GSK3008348, characterization of the pharmacology, the physico-chemical, and PK properties of the molecule were assessed to track suitability for inhaled delivery. Most of these properties were evident in the GSK3008348 molecule except for lung retention[27]. Therefore, the duration of

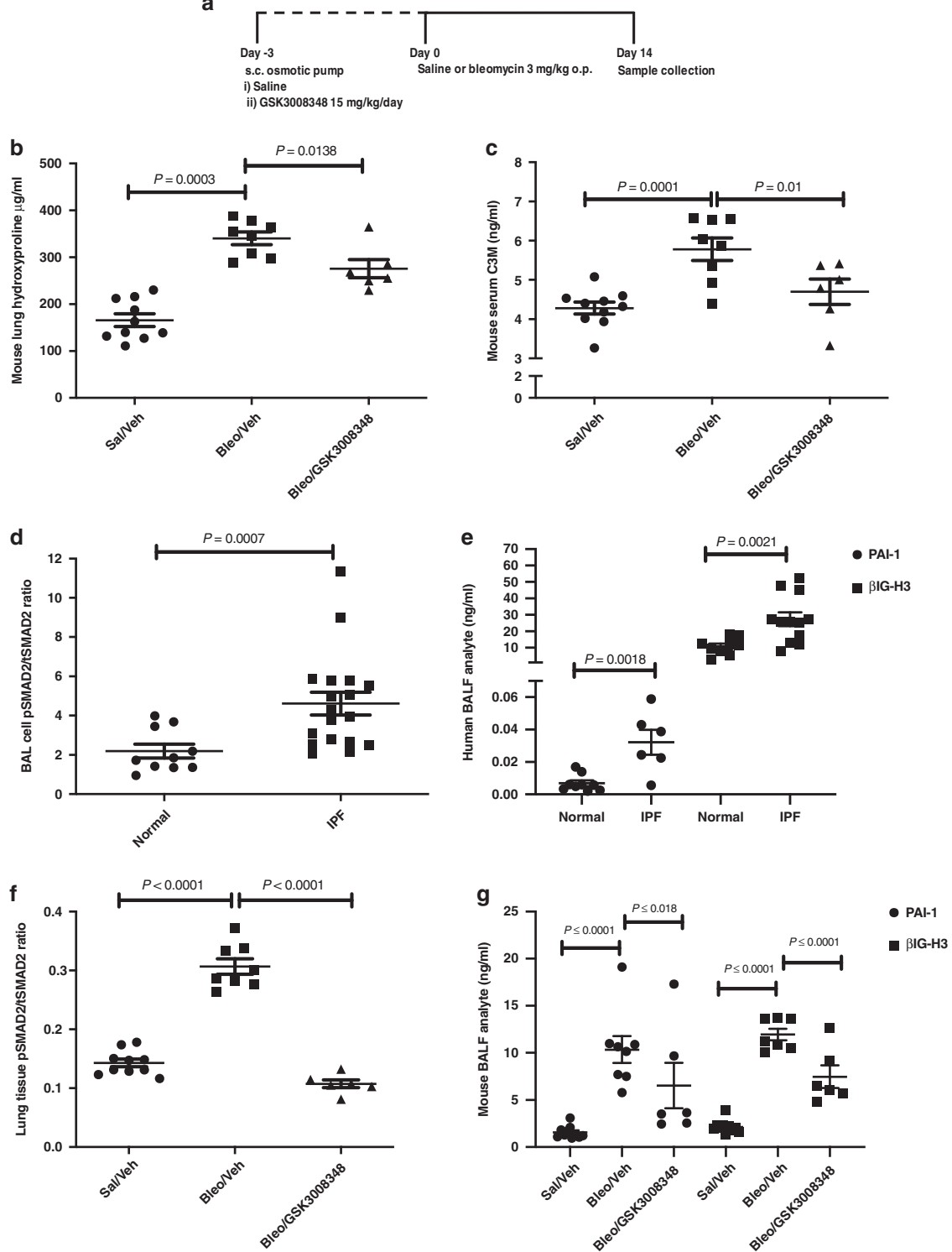

**Fig. 6 Translational PD and disease biomarkers from pre-clinical to clinical samples. a** Mice were treated with GSK3008348 prior to bleomycin challenge and the effect of GSK3008348 on the **b** collagen biomarkers hydroxyproline and **c** C3M in lung tissue (mean ± SEM; n = 6–10 animals per group). Levels of pSmad2 in **d** BAL cells and **e** βIG-H3, PAI-1 BAL fluid **d** in normal and IPF patient populations (mean ± SEM; 6–18 donors). The effect of GSK3008348 on PD biomarkers of TGFβ-activation in **f** lung tissue (pSmad2) and **g** BALF (βIG-H3 and PAI-1) in bleomycin-treated mice (mean ± SEM; n = 6–10 animals per group). All statistical analysis completed by ANOVA with Fisher's LSD post test. *Sal/Veh* Saline/Vehicle, *Bleo/Veh* Bleomycin/Vehicle. Source data are provided as a Source Data file.

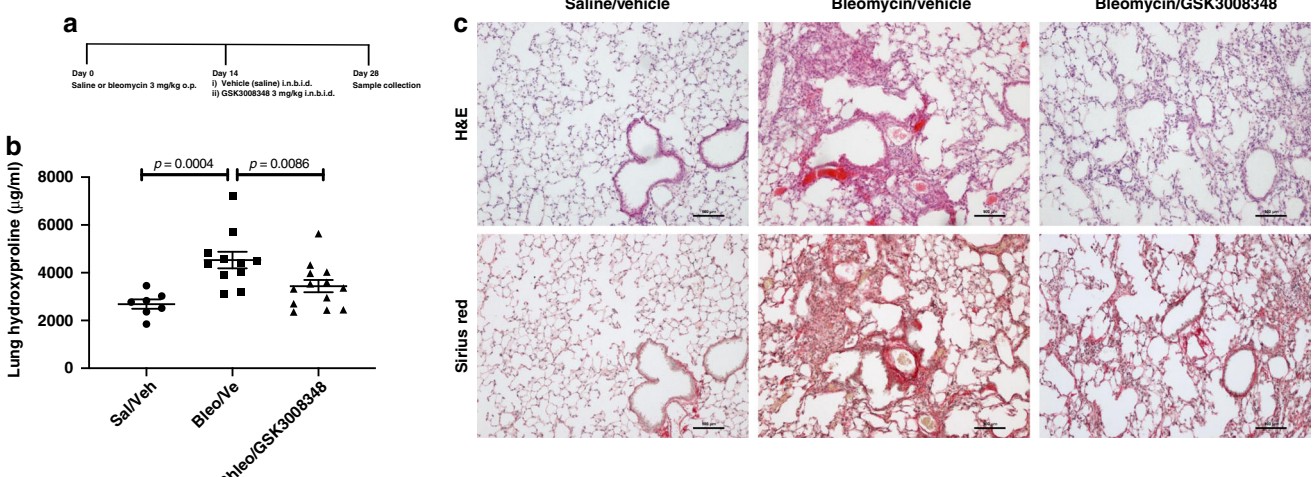

**Fig. 7 Effect of therapeutic dosing with GSK3008348 on lung collagen. a** Therapeutic dosing protocol. **b** Levels of collagen biomarker hydroxyproline in lung tissue from saline or bleomycin-treated mice in response to i.n. dosing with vehicle or GSK3008348 (mean ± SEM; $n = 7$–13 animals per group). **c** H&E and Sirius Red staining in representative lung sections from saline or bleomycin-treated mice ($n = 4$ animals per group). Scale bars are 100 μm. Statistical analysis completed by ANOVA with Fisher's LSD post test. b.i.d., twice daily; *Sal/Veh* Saline/Vehicle, *Bleo/Veh* Bleomycin/Vehicle. Source data are provided as a Source Data file.

action of this inhaled molecule was designed to be driven by the PD of the interaction between GSK3008348 and the αvβ6 integrin.

GSK3008348 demonstrates a high affinity for the αvβ6 integrin in soluble protein preparations but more importantly in plasma membranes generated from fibrotic regions of human IPF lung. The use of a radiolabeled version of GSK3008348 has enabled expression levels of αvβ6 integrins to be accurately quantified in normal and IPF lung. First, this has confirmed and quantified the upregulation of αvβ6 integrins in IPF previously observed semi-quantitatively using immunohistochemistry[12]. Second, the amount of integrin present in disease tissue can be incorporated into more accurate models to predict the therapeutic dose of αvβ6 integrin inhibitors. NanoSPECT-CT imaging studies using A20FMDV2 in saline and bleomycin-treated mice demonstrated the dose- and time-dependent engagement of GSK3008348 with αvβ6 in an in vivo model of lung fibrosis. The quantification of αvβ6 integrin upregulation in IPF and the profiling of GSK3008348 binding to the integrin in diseased tissue will allow this pre-clinical work to be translated to clinical studies in IPF[25,28].

It has been demonstrated for RGD-peptides that binding to αvβ6 integrin induces internalization, followed by a delay in the return of the integrin to the cell surface[29,30]. In this study, GSK3008348-induced rapid internalization of the αvβ6 integrin that was concentration-, time-, and clathrin-dependent, in agreement with published data[31]. Following washout of GSK3008348, the return of αvβ6 integrins to the cell surface was markedly slower compared with endogenous turnover[31] as well as post internalization via RGD-peptides[30]. Furthermore, in NHBE cells in the constant presence of GSK3008348 over 24 h, a complete loss of αvβ6 integrins was also observed at both the cell surface and intracellularly. Both these observations suggest that the internalized GSK3008348–αvβ6 complex is degraded and that returning αvβ6 integrins in washout studies are newly synthesized. Furthermore, in washout experiments investigating αvβ6-mediated inhibition of TGFβ activation, the sustained inhibition observed with GSK3008348 was prevented in the presence of chloroquine. These observations suggest that GSK3008348 induces internalization of αvβ6 integrins that are subsequently sorted for degradation in lysosomes. The high affinity of

GSK3008348 for the αvβ6 integrin combined with its previously described fast association and slow dissociation profile[32] likely results in a prolonged activation of the integrin in intracellular vesicles. We hypothesize that the longer the integrin is engaged at the RGD-site post internalization, the more likely it is to be intracellularly designated for degradation. This effect has previously been observed with other small molecule/protein interactions[33,34] and would ultimately result in the prolonged duration of inhibition of TGFβ activation in vivo demonstrated in this study.

One of the current challenges of clinical study design within the IPF field is the size and duration of studies required to measure meaningful changes in lung function or mortality, especially in addition to standard of care[4–7]. Therefore, there have been significant efforts to identify IPF biomarkers detectable in blood or BALF samples that reflect disease biology and which could be used as early surrogates for long-term decline in forced vital capacity or mortality, predict disease progression and allow enrichment of patient cohorts for clinical studies as well as facilitating potentially shorter and smaller studies.

The purpose of the bleomycin model of fibrosis in these studies was twofold. Initial studies were designed primarily to assess target engagement and effects on TGFβ signaling and other PD biomarkers rather than to predict clinical efficacy. Therefore, these studies used a low bleomycin dose (1 mg/kg) to induce a mild bleomycin injury and minimize distress to animals where possible. A higher dose of bleomycin (3 mg/kg) was used in studies in which fibrotic end points (hydroxyproline) were assessed including both the prophylactic and therapeutic dosing studies with GSK3008348. We demonstrated that prophylactic treatment with GSK3008348 led to a modest inhibition of lung hydroxyproline similar to values reported following prophylactic dosing with nintedanib in bleomycin-treated mice[35]. Subsequent studies were used to assess the feasibility of inhaled small molecules to penetrate and ameliorate established lung fibrosis, and indeed 2 weeks inhaled GSK3008348 lead to a substantial reduction in established fibrosis as measured by total lung collagen and histological assessment. However, there are known limitations of the bleomycin model of pulmonary fibrosis that mean it cannot reliably predict the clinical effectiveness of an anti-fibrotic therapy[36]. Therefore, to ensure the translational

efficacy of GSK3008348 could be determined, we also measured the effects of GSK3008348 on established and recently identified biomarkers each of which could be translated rapidly into clinical practice. Initial studies evaluated the effects of GSK3008348 on TGFβ activation pathways. These data established that levels of pSmad2 measured in BAL cells recovered from the fibrotic lungs of bleomycin-treated mice were reduced following administration of GSK3008348 in comparison with vehicle control treated mice, thus demonstrating engagement of the mechanism in the lung. Furthermore, proteins known to be regulated by active TGFβ including PAI-1 and βIG-H3, were also inhibited by GSK3008348 in mouse BALF confirming the effects of GSK3008348 on the TGFβ activation pathway and offering further options for measuring a PD effect on this signaling pathway in clinical studies. Moreover, we demonstrated that GSK3008348 could inhibit pSmad2 in human diseased samples using PCLS from IPF patients confirming that GSK3008348 can inhibit TGFβ activation pathways both in human fibrotic tissue as well as a murine model of lung fibrosis. These data suggest that measuring levels of pSmad2 in BAL cells obtained from clinical studies in the presence of GSK3008348 could be a good surrogate measure of the effects of this αvβ6 inhibitor on TGFβ activation in IPF lung. One promising and recently identified serum biomarker in IPF is the MMP-degraded ECM protein neoepitope C3M, which is also raised in the bleomycin model of lung fibrosis[37] and is associated with disease progression and mortality in patients with IPF[23]. We demonstrated that GSK3008348 was able to inhibit serum C3M in the murine model of bleomycin-induced fibrosis and may therefore be useful as a mechanism-specific biomarker in future IPF clinical trials.

Although αvβ6 integrins are considered to drive the majority of TGFβ activation in IPF, there are data suggesting a role for αvβ5[38] and, more recently, αvβ1[19]. However, data generated in this study utilizing PCLS from diseased human IPF tissue has compared the effects of GSK3008348 with highly selective αvβ1 (c8) and αvβ5 inhibitors and shown that of these the αvβ6 integrin appears to be the major contributor to TGFβ activation and downstream phosphorylation of Smad2 in this system. The comparable potency for inhibition of pSmad2 in primary cells and in human lung tissue suggests that the epithelial αvβ6 integrin is the main driver of TGFβ activation in human IPF tissue. However, this does not discount a potential role of other integrins in the pathogenesis of pulmonary fibrosis. Although epithelial cell injury may be a driver of pulmonary fibrosis, myofibroblasts are a key effector cell and fibroblast to myofibroblast differentiation is a major step in this process. Active TGFβ is well known to induce myofibroblast differentiation, which is mediated in part through Smad signaling, but also by non-canonical activation of focal adhesion kinase (FAK) through interactions with integrins[39]. Therefore, even if epithelial cells do generate sufficient active TGFβ to lead to downstream TGFβ signaling in neighboring fibroblasts, as these data suggest, it is possible that myofibroblasts develop autonomous, non-canonical, pro-fibrotic pathways through TGFβ induced αvβ1 integrin and FAK activation that will not be inhibited by GSK3008348. Therefore, further studies are required to determine the relative contribution of αvβ1 and αvβ6 integrins on myofibroblast differentiation.

This study has some strengths and limitations that will only be fully addressed in clinical studies with GSK3008348. A key strength of this study is the demonstration that twice daily (b.i.d.) inhaled dosing can ameliorate established bleomycin-induced pulmonary fibrosis. This is especially important in light of the recent early termination of the STX100 study owing to safety concerns. Although it is not known at the current time whether the adverse effects of STX100 were due to on-target effects, the difference in drug delivery (i.n. versus s.c.), dosing schedule (b.i.d.

versus weekly) and TGFβ inhibition (reduced rather than blocked completely) offer considerable advantages for GSK3008348 compared with the αvβ6 antibody blocking approach. The additional strengths of this study are the comprehensive assessment of both the anti-fibrotic potential and target engagement of the molecule using both human and murine model systems in addition to measuring both traditional biochemical and clinically translatable end points. Limitations include use of the bleomycin mouse model of lung fibrosis that is recognized to be a poor surrogate for clinical efficacy in IPF. However, the use of the bleomycin model in this study has been to demonstrate that inhaled delivery of an integrin inhibitor can mitigate established fibrosis and to establish PD and mechanism-specific biomarkers. These data demonstrate that inhalation of a small molecule can penetrate fibrotic lung and the biomarkers studies demonstrate proof of mechanism that can be further assessed in future IPF clinical trials. Thus, these data have been used to create a clear clinical translational pathway and to generate and test a hypothesis, rather than to raise unrealistic hopes for clinical efficacy at this stage of pre-clinical development.

All the data generated as part of this study have contributed to the design and clinical dose prediction used for GSK3008348 in phase Ib studies where the ability of this molecule to engage the αvβ6 integrin in IPF patients is currently under investigation[25]. Based on this translational data package the hypothesized mechanism of action of GSK3008348 post administration into an IPF patient's lung is as follows: (1) nebulized GSK3008348 would enter the lung and bind to αvβ6 with high affinity and fast association kinetics. (2) The GSK3008348/αvβ6 complex would then be rapidly internalized in minutes and the integrin degraded in lysosomes. (3) The αvβ6-mediated TGFβ activation in the IPF lung would be inhibited reducing pro-fibrotic mediators and extracellular matrix deposition. These studies therefore represent a different approach to pre-clinical drug development that will reduce the chances of failed translation to early phase clinical studies and increase the prospect of generating effective therapies for IPF.

## Methods

**Study design**. The objective of this study was to develop a clear path for a proposed IPF therapeutic from pre-clinical to clinical studies that linked target engagement through to functional inhibition of PD and disease end points. In order to achieve this, studies focused on human primary cells and IPF disease tissue in in vitro and ex vivo studies to characterize the pharmacology of an αvβ6 RGD-mimetic inhibitor investigating binding, receptor internalization and inhibition of TGFβ. Studies investigating these end points in vivo were then completed using a mouse model of pulmonary fibrosis, with disease end point relevance and validation determined in ex vivo BAL cell and fluid from IPF patients that could be readily profiled in future clinical experiments.

All animal studies were approved by the University of Nottingham Animal Welfare and Ethical Review Board and carried out in accordance with Animals (Scientific Procedures) Act 1986, the GSK Policy on the Care, Welfare and Treatment of Animals and the ARRIVE (Animal Research: Reporting of In Vivo Experiments) guidelines[40]. Mice were randomized into treatment groups and for initial studies sample size was selected based on experience of the model. In subsequent studies, power analysis was completed on the previous studies data to determine sample size.

The human biological samples were sourced ethically, and their use in research use complied with the terms of the written informed consents and under an IRB/EC approved ethics protocol at each institution. Normal (healthy non-fibrotic) human lungs (designated by medical history) from organ donors were obtained from the National Disease Research Interchange (Philadelphia, PA, USA) in accordance with local human biological sample management procedures. Lung tissue samples from patients with IPF undergoing lung transplantation were obtained from the Institute of Transplantation at Newcastle Upon Tyne Hospitals. All patients had their IPF diagnosis confirmed based on their medical history and evaluation of their explanted lung tissue by a board-certified respiratory pathologist. Tissue was obtained in accordance with local human biological sample management procedures which was approved by regional ethics approval (11/NE/0291). BAL samples were obtained from IPF patient at the National Institute for Health Research Respiratory Biomedical Research Unit (Royal Brompton Hospital, London, UK) as part of routine diagnostic BAL in the PROFILE study[23].

All chemicals and reagents were purchased from Sigma-Aldrich Co. Ltd. (Gillingham, Kent, UK) and antibodies from R&D Systems (Minneapolis, MN, USA) unless otherwise stated. All tissue culture flasks and plates were purchased from Greiner Bio-One (Firckenhausen, Germany). All in vitro studies were completed at least three times and at 37 °C, unless otherwise stated. Cellular assays were completed in an atmosphere of 5% $CO_2$ with a relative humidity of 95% at 37 °C.

**Synthesis of inhibitors**. The small molecule compounds GSK3008348[21], c8 (αvβ1 inhibitor[19]), SB-525334 (TGFβR1 inhibitor[41]), and SB-267268 (αvβ3/αvβ5 inhibitor[42]) used in this study were synthesized by the Fibrosis DPU Medicinal Chemistry group at GlaxoSmithKline Medicines Research Centre (Stevenage, Hertfordshire, UK). The αvβ6 selective peptide A20FMDV2[43] was synthesized by Cambridge Research Biochemicals (Cleveland, UK).

**Radioligand binding assays**. Radioligand binding and platelet aggregation (αIIbβ3 only) assays were completed against the RGD integrins. All radioligand binding experiments were performed in 96-deep well plates at 37 °C in binding buffer (25 mM 4-(2-hydroxyethyl)-1-piperazineethanesulfonic acid, 100 mM NaCl, 2 mM $MgCl_2$ and 1 mM 3-[(3-cholamidopropyl)dime thylammonio]-1-propanesulfonate at pH 7.4 (NaOH)) in a total volume of either 0.5 ml or 1.5 ml (1.5 ml for saturation binding studies with lung parenchyma membranes) consisting of 50 μl/ well of either unlabeled compound at varying concentrations or vehicle (1% dimethyl sulphoxide; DMSO), 50 μl of [³H]RGD ligand[32] or [³H]GSK3008348 and either 0.4 or 1.4 ml/well of purified integrin or membranes (concentration dependent on the number of binding sites of individual soluble protein or membrane fragment preparations). Non-specific binding (NSB) was determined with 10 μM SC-68448 (pan-αv RGD small molecule[32]) except for saturation binding experiments with lung membranes where 10 μM A20FMDV2 (selective αvβ6 RGD-peptide) was used. Specific binding was measured by subtracting the NSB from the total radioligand binding in the presence of vehicle (1% DMSO). Binding was terminated by rapid vacuum filtration and the amount of specific radioligand bound was measured by LS spectroscopy using a TriCarb 2900 TR LS counter (PerkinElmer LAS UK Ltd., Beaconsfield, UK). Binding assays were completed in the presence of 2 mM $Mg^{2+}$ to standardize integrin activation state.

For platelet aggregation, platelets were resuspended in assay buffer (5 mM HEPES, 140 mM NaCl, 3 mM KCl, 12 mM $NaHCO_3$, 7 mM $KH_2PO_4$, and 60 mM D-glucose at pH 7.4 (NaOH)), and resuspended to 4 × 108 cells/ml. Cells were then incubated for 30 min at ambient temperature (20–22 °C) with the addition of 0.75 mM $MgCl_2$, 1.5 mM $CaCl_2$, and 0.4 mg/ml fibrinogen and added to clear 96-well flat bottomed plate (90 μl/well) containing 10 μl/well vehicle (0.1% DMSO in platelet aggregation assay buffer) or test compounds. Plates were then sealed and incubated for 20 min at 37 °C prior to a basal absorbance read where the optical density of each well was read on a SpectraMax Plus 384 (Molecular Devices, Sunnyvale, CA, USA) set to 450 nm. Following the addition of 10 μl/well adenosine diphosphate the optical density of each well was read at 450 nm every 20 s over a 6 min kinetic time course read with the basal absorbance read used to measure the % inhibition of platelet aggregation over the kinetic time course read.

**Cell culture**. NHBE were purchased from Lonza (Lonza Group Ltd, Basel, Switzerland) and placed into culture as a monolayer adhered in type I collagen coated tissue culture flasks using aseptic techniques. Cells were maintained in NHBE cell medium (bronchial epithelial growth medium (BEGM) containing 0.6 mM/l MgCl) supplemented with BEGM CloneticsTM SingleQuots (containing bovine pituitary extract, insulin, hydrocortisone, GA-1000 (consisting of 30 mg/ml gentamicin and 15 μg/ml amphotericin), retinoic acid, transferrin, triiodothyronine, epinephrine, and human epidermal growth factor)) in 95%:5% air:CO₂ and were harvested when ~80% confluent using Accutase®. Cells were then resuspended in phosphate-buffered saline (PBS), centrifuged at 300 × g for 5 min prior to re-suspension in either NHBE cell medium or flow cytometry buffer (RPMI 1640 (without L-glutamine and phenol red) containing 10 mM/l HEPES, 1% w/v bovine serum albumin (BSA) and 2 mM/l MgCl₂.

Cultures of SAEC from explanted IPF lung tissue were established from brushings collected from airways of <2 mm in diameter. Cells were cultured in Small Airway Growth Media (Promocell) and used at P4 or less in experiments. SAEC exhibit the typical cobblestone appearance of an epithelial cell monolayer in culture, express epithelial markers (Cytokeratin-17 and E-cadherin) but are negative for mesenchymal (fibronectin), leukocyte (CD45), alveolar macrophage (CD68), and endothelial (CD31/PECAM) markers. Cells were lifted from flasks with Trysin ethylenediaminetetraacetic acid (EDTA) solution, washed in complete media, centrifuged at 300 × g for 5 min prior and resuspended in flow cytometry buffer for analysis.

The epithelial transformed mink lung cell (TMLC) line that expresses firefly luciferase under the control of a TGFβ-sensitive portion of the PAI-1 promoter[44] was obtained from Professor Daniel Rifkin (New York University, NY, USA) and maintained in culture in DMEM containing 10% fetal bovine serum with 2 mM L-glutamine, penicillin/streptomycin, and 200 μg/ml G418. Cells were lifted from flasks with 5 mM EDTA, washed in PBS with 0.5% BSA, counted by hemocytometer, and plated in 96-well plates.

**Flow cytometry and imaging assays**. The measurement of surface and intracellular αvβ6 in NHBE cells and SAEC cells was determined via flow cytometry. NHBE flow cytometry assays were performed in 96-well polypropylene microplates with NHBE cells suspended in flow cytometry buffer (45 μl/well with 70,000 cells/well) in the presence of appropriate concentrations of drug or vehicle (0.1% DMSO) as required. Where stated, cells were permeabilised by incubating with 0.2% w/v saponin for 5 min at ambient temperature (20–22 °C) prior to compound/vehicle addition. Inhibition of clathrin or lipid raft mediated endocytosis was investigated by pre-incubating NHBE cells with either 2 μg/ml chlorpromazine or 10 μg/ml filipin, respectively, for 5 min prior to addition of compound or vehicle (0.1% DMSO). SAEC flow cytometry assays were performed in flow cytometry tubes with 100,000 cells in a 100 μl volume of flow cytometry buffer in the presence of GSK3008348 or vehicle (0.1% DMSO). All experiments were stopped by addition of 10 μl per well/tube of IgG2B-PE or hβ6-PE and incubation at 4 °C for 1 h. Cells were then washed twice by centrifuging at 300 × g for 5 min, removing supernatant and adding flow cytometry buffer (150 μl/well). After the second wash cells were resuspended in flow cytometry buffer (200 μl/well) and cell suspensions transferred into a 96-well round bottom polypropylene plate (Corning Inc. Life Sciences, Tewksbury, MA, USA) or flow cytometry tubes. Cell samples were then acquired on a fluorescence-activated cell sorting (FACS) Canto II (BD Biosciences, San Jose, CA, USA) using a high throughput sampler system and BD FACS DivaTM version 6.1.3 software. Cells were identified by their forward and side-scatter characteristics and a single population gated with cell debris excluded (Supplementary Fig. 3) and the mean fluorescence intensity (MFI) of antibody conjugated cells measured. The fluorescence was quantified on at least 5000 cells and following acquisition all data was exported as flow cytometry standard format 3.0 files with raw data values captured as MFI. Fluorescence-activated cell analyses histograms were plotted using FlowJo software (Tree Star Inc. Ashland, OR, USA).

For internalization, concentration–response curves NHBE cells were added to 96-well polypropylene microplates containing 5 μl/well compound at varying concentrations or vehicle 0.1% DMSO. Plates were incubated for 2 h in 95%:5% air:CO₂ at 37 °C prior to addition of 10 μl IgG2B-PE or hβ6-PE antibody. For determining the rate of ligand-induced αvβ6 internalization NHBE cells were added to 96-well polypropylene microplates containing 5 μl/well compound (at a concentration that caused maximal internalization) or vehicle (0.1% DMSO). Plates were incubated in 95%:5% air:CO₂ at 37 °C for varying times up to 1 h and then transferred immediately on to ice to stop any further internalization. In all, 10 μl/well IgG2B-PE or hβ6-PE antibody were then added to plates. For determining the rate of αvβ6 return to the cell surface (reversal of ligand-induced αvβ6 internalization), NHBE cells were added to 96-well polypropylene microplates containing 5 μl/well compound (at a concentration that caused maximal internalization) or vehicle (0.1% DMSO). Plates were incubated in 95%:5% air:CO₂ at 37 °C for 1 h then centrifuged at 500 × g for 5 min, supernatant removed and cell pellets resuspended in PBS (150 μl/well). This process was repeated prior to re-suspension of cells in cell medium (150 μl/well) and incubation for varying times up to 48 h. Plates were then transferred on to ice and 10 μl/well IgG2B-PE or hβ6-PE antibody added. All samples were then processed and read on the FACS Canto II as detailed above.

For cell imaging studies, NHBE cells were adhered in glass chamber slides and incubated with vehicle (0.1% DMSO) or GSK3008348 in the absence or presence of 10 μM chloroquine for 2 h prior to β6 integrin staining with sheep anti-human integrin β6 antibody then donkey anti-sheep IgG Alexa Fluor® 488 antibody (Invitrogen Ltd., Renfrewshire, UK). Chamber slides were imaged with a Leica TCS SP5 confocal microscope (Leica Microsystems Inc., Buckinghamshire, UK) to produce a 3D cell z-stack. Quantitative analysis of the intensity of β6 staining was completed using Columbus™ Image Data Storage and Analysis System (PerkinElmer LAS UK Ltd., Buckinghamshire, UK). For high-content screening (HCS) studies, supplement starved NHBE cells were plated in collagen I coated 96-well imaging plates and treated with vehicle (0.1% DMSO) or GSK3008348 in the presence or absence of 10 μM chloroquine for 24 h prior to β6 integrin staining with mouse anti-human integrin β6 then goat anti-mouse IgG Alexa Fluor® 488 antibody (Invitrogen Ltd., Renfrewshire, UK). Image acquisition and analysis of stained cells were performed using the Arrayscan VTI High Content Reader (Thermo Fisher Scientific, MA, USA), applying bioapplications to quantify total β6 immunofluorescence. For all imaging studies, nuclear staining was completed with Hoechst 33342 dye (Invitrogen Ltd., Renfrewshire, UK) and if required cells were permeabilised with 0.2% w/v saponin (flow cytometry) or Triton X-100 (HCS) to determine total αvβ6 staining (membrane and intracellular).

**TGFβ activation and signaling assays**. To determine activation of TGFβ via αvβ6 in NHBE cells following washout, a TMLC co-culture system was used. Cultured NHBE cells were harvested, resuspended in NHBE cell medium and 25,000 cells (100 μl/well) seeded in 96-well clear collagen I coated plates then left for 24 h to adhere in 95%:5% air:CO₂ at 37 °C before removal of medium. Cultured TMLCs were harvested, resuspended in TMLC co-culture medium containing 20 μM lysophosphatidic acid (LPA) and 25,000 cells (100 μl/well) seeded onto the NHBE cells. Vehicle (0.1% DMSO) or test compounds were then added (10 μl/well) and plates incubated for 24 h in 95%:5% air:CO₂ at 37 °C. Supernatants were then removed from the NHBE cell/TMLC co-culture and 100 μl/well PBS (containing 1 mM CaCl₂ and 1 mM MgCl₂) and 100 μl/well Steady-Glo® Reagent (Promega

Corporation, Madison, WI, USA) added. Plates were then incubated at ambient temperature (20–22 °C) for 5 min before supernatants were transferred to 96-well white, solid bottom plates and luminescence read on a MicroBeta® TriLux (Per-kinElmer LAS UK Ltd., Beaconsfield, UK). For washout studies, vehicle (0.1% DMSO) or GSK3008348 in the absence or presence of 10 μM chloroquine were incubated with the NHBE cell/TMLC co-culture for 1 h prior to washout. Washout studies were also completed in the presence of chloroquine (10 μM made up in TMLC co-culture medium). TGFβ was quantified by comparing values obtained under experimental conditions to readings obtained from a standard curve derived from increasing concentrations of active TGFβ added to the co-culture under identical conditions.

TGFβ levels in supernatants harvested from human IPF PCLSs were determined by incubation with TMLC cells for 18 h. For the measurement of concentration-dependent effects of αvβ6 and TGFβR1 inhibitors on TGFβ activation and signaling in NHBE cells, pSmad2 (normalized to GAPDH) was measured. pSmad2 and GAPDH was quantified using the Milliplex TGFβ Signaling Magnetic Bead Panel 6 plex (Merck Millipore, Billerica, MA, USA) luminex assay according to the manufacturer's instructions. pSmad2 was determined using the same method for in vivo and ex vivo tissue and BAL samples, however data were normalized to tSmad2. tSmad2 was measured in the Pathscan Total Smad2 sandwich ELISA kit (Cell Signaling Technologies, London, UK), according to the manufacturer's instructions with standard curves generated using recombinant GST-Smad2. βIG-H3 was determined in mouse and human BAL fluid samples using the mouse or human βIG-H3 DuoSet ELISA kit (R&D Systems, Abingdon, USA) respectively, according to the manufacturer's instructions. PAI-1 was determined in mouse and human BAL fluid samples using the mouse or human PAI-1 Total antigen ELISA kit (Molecular Innovations, Novi, MI, USA) respectively, according to the manufacturer's instructions.

**In vivo mouse studies**. Male C57BL/6 mice (6–12 weeks old from Charles River, Kent, UK) were acclimatized for 5–7 days before undergoing procedures and ranged from 16 to 28 g at the time of study. Animal welfare followed the requirements of the Home Office Code of Practice for the Housing and Care of Animals used in Scientific Procedures. In brief, mice were housed in individually ventilated cages in groups of three to four in a temperature and humidity controlled environment, with a 12 h light:dark cycle with food and water available ad libitum. Play tunnels and bedding material were used to provide environmental enrichment. In addition to the standard weight and health checks performed by researchers, animals were monitored at least once daily by a trained animal technician to ensure that they were in good health. Mice were randomized between the different treatment groups for each experiment and were observed regularly during the study. Mice used were either naive or treated with a single oropharyngeal 50 μl dose (1 mg/kg or 3 mg/kg) of bleomycin sulfate (Bleo-Kyowa; Aesica Queenborough Ltd, Kent, UK) or saline vehicle.

In prophylactic studies s.c. osmotic mini-pumps (ALZET® mini-osmotic pump model 2004) were surgically implanted 3 days prior to bleomycin dosing and GSK3008348 in sterile saline was administered at a dose of 15 mg/kg/day for 14 days post bleomycin instillation. For therapeutic dosing studies, mice were dosed twice daily by i.n. inhalation of 3 mg/kg GSK3008348 from day 14 to day 28 post bleomycin instillation (3 mg/kg).

In bleomycin studies assessing the effect of single doses of compound on PD biomarkers, 1 mg/kg GSK3008348 in sterile saline was administered via the i.n. route (50 μl administered equally between nostrils to mimic its intended clinical dose route) either 7, 14, or 28 days post bleomycin instillation. In single-dosing studies measuring Smad2 levels in lung tissue and BAL cells, SB-525334 (TGFβR1 inhibitor) was dosed via oral gavage at 30 mg/kg (0.2% (v/v) Tween 80 in saline, adjusted to pH 4.1 with NaOH) at day 14 post bleomycin instillation, with sample collection completed 2 h post dosing.

At the end of studies, mice were culled by an overdose of pentobarbitone sodium (Euthatal; Merial Animal Health Ltd, Essex, UK) and death confirmed by the permanent cessation of circulation and/or dislocation of the neck as appropriate. At the defined time points post dosing detailed in the "Results" section, the following samples were collected as applicable to each study. Where BAL cells and BALF were required, samples were collected by slowly aspirating and withdrawing 0.5 ml PBS into the trachea using a flexible butterfly catheter. This was repeated three times and BAL fluid (total ~1.5 ml) was pooled, treated with 15 μL protease and phosphatase inhibitors and kept on ice. An aliquot of BAL fluid was used to perform a total cell count. The remaining BAL fluid was centrifuged (150 × g for 10 min at 4 °C) and the cell pellet and BAL supernatant stored separately at −80 °C. Lungs were removed, snap frozen, and ground to powder under liquid nitrogen, divided into two aliquots and processed for measurement of pSmad2 and/ or hydroxyproline. In animals that had undergone BAL sample collection, the lung samples for hydroxyproline analysis was dried at 60 °C for 2 h prior to processing otherwise the lung samples were processed for the analysis of hydroxyproline levels. For hydroxyproline assays, the lung powder was mixed in 1 ml distilled water on ice in Pyrex tubes and incubated in 125 μl 50% TCA (Sigma-Aldrich, Gillingham, Kent, UK) at 4 °C for 20 min. Samples were centrifuged at 528 × g for 10 min at 4 °C, hydrolyzed in 1 ml of 12 N HCL overnight at 110 °C prior to reconstitution in 2 ml of distilled water. In total, 200 μl of hydrozylate, or hydroxyproline standard, were oxidized in 500 μl chloramine T for 20 min at room

temperature (20–22 °C) prior to the addition of 500 μl of Ehrlich's solution and incubation at 65 °C for 15 min. Samples were then incubated at room temperature for 2 h prior to colorimetric analysis at an absorbance of 550 nm. Hydroxyproline per mg of lung tissue was calculated against a standard curve.

The remaining aliquot of lung powder was lysed in Cell Lysis Buffer (Cell Signaling Technology, MA, USA) containing protease and phosphatase inhibitors (Halt™ protease and phosphatase inhibitors cocktail (Thermo Fisher Scientific, MA, USA)) according to the manufacturer's instructions. For animals requiring histological end point analysis, lungs were insufflated with 10% formalin (VWR chemicals, UK) at constant gravitational pressure (20 cm H2O) then paraffin wax embedded for histology.

Where applicable, a blood sample from each animal was taken via cardiac puncture and split between containers with or without anti-coagulant (lithium heparin) to permit both for PK and PD analysis. The concentration of parent compound in blood was determined by reverse phase HPLC/MS/MS using an electrospray ionization interface in positive ion mode against a matrix matched calibration line. The lower limit of quantitation in blood of GSK3008348 was 2 ng/ ml. The serum levels of the MMP-degraded ECM protein C3M was determined in a competitive ELISA-based, neoepitope assay manufactured by Nordic Bioscience (Herlev, Denmark) and performed according to the manufacturer's specifications. In brief, 96-well pre-coated streptavidin plates were coated with biotinylated peptides specific for the protein of interest and incubated for 30 min at 20 °C. A volume of 20 μl of standard peptide or pre-diluted serum sample were added followed by addition of peroxidase-conjugated monoclonal antibodies and incubated for 1 h at 20 °C. Next, tetramethylbenzinidine (TMB) (Kem-En-Tec Diagnostics, Denmark) was added and the plates were incubated for 15 min at 20 ° C. All incubations included shaking of the plates followed by five washes with buffer (20 mM Tris, 50 mM NaCl, pH 7.2). The reaction of TMB was stopped by adding 1% sulfuric acid and the absorbance was measured at 450 nm, with 650 nm as reference. A standard curve was plotted using a four-parametric mathematical fit model and data were analyzed using the Softmax Pro (version 6.3) software. Levels of the biomarkers were measured in duplicates.

Mice undergoing nanoSPECT-CT imaging were injected intravenously via the tail vein with 15–30 MBq of [111In]-labeled DTPA-A20FMDV2 peptide in sterile water using a 50 μl dose volume 1 h prior to scanning. Mice were anaesthetized by exposure to isofluorane (3%), transferred to the sealed CT/SPECT tube and maintained under anesthesia at 1% isofluorane for the duration of the scan (45 min in total) at an oxygen flow rate of 1 L/min. Whole-body helical CT and SPECT scans were performed with a nanoSPECT-CT imaging system (Bioscan Inc, Washington, DC) fitted with four tungsten collimators and nine 1.4 mm diameter pinholes. SPECT images were obtained 2 h (saline/GSK3008348) or 24 h (IgG isotype control/10D5 anti-αvβ6 antibody intra-peritoneal administration in PBS at 2.5 mg/kg (Merck Millipore, Billerica, MA, USA)) post dosing of test agents with a time per view of 60 s resulting in a scan time of 30–45 min. CT images were obtained using a tube voltage of 45 kVp and an exposure time of 500 ms per view[22].

**Lung histology**. Histological sections of murine lung were cut at three microns and dewaxed in xylene prior to rehydration in decreasing concentrations of ethanol. The sections were incubated in either Mayer's haemotoxylin, and eosin or Sirius red and Weigert's haemotoxylin. Tissue staining was imaged using Nikon Eclipse 90i microscope and NIS Elements AR3.2 software (Nikon).

**Ex vivo human studies**. Human fibrotic lung tissue was collected from human explants after transplantation. Tissue which the pathologist deemed suitable for research was inflated with 2–3% low boiling point agarose, with the agarose being allowed to set at 4 °C. PCLS were then cut at 400 μm on a vibrating microtome and cultured in DMEM media. PCLS were rested for 48 h prior to treatment. PCLS were incubated with inhibitors for 48 h prior to tissue slices being lysed in 1x Cell Lysis Buffer (Cell Signaling Technology, MA, USA) containing protease and phosphatase inhibitors (Halt™ protease and phosphatase inhibitors cocktail (Thermo Fisher Scientific, MA, USA)) according to the manufacturer's instructions.

BAL cell and fluid samples were obtained from patients undergoing routine diagnostic BAL. BALF cell pellets were prepared by centrifugation (300 × g for 5 min) and differentially counted. Following re-suspension cells were incubated in RPMI containing 0.1% DMSO (vehicle control) or inhibitor (30 min at 37 °C in 95%:5% air:CO2). Following incubation, suspensions were centrifuged (300 g for 5 min) and cell pellets lysed in ice cold PhosphoSafe® buffer (Merck Millipore) before being assayed for pSmad2 and total Smad.

**Statistical analysis**. Statistical analyses were completed using GraphPad Prism 7.0 (GraphPad Software, San Diego, CA, USA) for in vitro studies and R version 3.4 for in vivo/ex vivo studies. Statistical significance between two data sets was tested using a Student's unpaired $t$ test. One-way analysis of variance was used for comparison of more than two data sets and, where significance was observed, an appropriate post test completed. Unless otherwise indicated data shown graphically, and in the text, are either mean ± standard deviation or, where three or more data points/individual experiments have been completed, mean ± standard error of the mean.

**Reporting summary**. Further information on research design is available in the Nature Research Reporting Summary linked to this article.

## Data availability

The authors declare that the data supporting the findings of this study are available within the paper and its supplementary information files. Source data are provided with this paper.

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

## Acknowledgements

We would like to acknowledge the entire scientific and management team at GlaxoSmithKline that contributed to the discovery, characterization, and progression of GSK3008348. We thank S.B. Ludbrook, W.A. Fahy, A.D. Blanchard, P.A. Procopiou, P. Szeto, P. Saklatvala, D.C. Budd, D.A. Hall, J.C. Denyer, N.J. Shipley, and M.G. Lennon for their support and input into this drug discovery project. This work was fully supported by GlaxoSmithKline.

## Author contributions

R.J.S. wrote the paper and contributed to overall study design and data interpretation; A.E.J., R.G.J., S.J.F.M., and P.T.L. contributed to the writing of the paper. A.E.J. performed and contributed to in vivo study design and data interpretation; G.V., J.L.M., R.H.G., and J.W.B. performed or supervised DMPK studies; R.J.S., A.E.J., R.H.G., K.T.P., E.J.F., P.F.M., R.F.R., M.H., L.I.B., E.G., V.S.M., Y.M., J.A.R., J.C.L., L.A.B., B.S.B., R.A.B.,

R.B., J.L., D.J.F., S.P., A.H., L.A.O., C.J., R.C.E.P., N.S.G., D.J.L., R.C.C., and R.G.J. designed, performed, and/or analyzed in vitro and/or in vivo studies, or the supervision thereof; J.L. performed homology modeling; T.M.M. and A.J.F. provided human IPF BAL and lung tissue, respectively, and contributed to data interpretation. R.G.J., P.T.L., and R. P.M. contributed to overall study design and data interpretation.

## Competing interests

T.M.M., A.J.F., R.C.C., and R.G.J. are or have been paid consultants to GlaxoSmithKline. The remaining authors declare no competing interests.
