## [Peer Review File · Nature Communications]

Reviewers' comments:

Reviewer #1 (Remarks to the Author):

This study by John et al. reports on the development on a novel small molecule RGD-mimetic that inhibits integrin avb6 and, in doing so, diminishes TGF-b1 activation and impacts lung fibrosis, and the data presented could lay the pre-clinical foundation for the translation of this compound, which is inhalationally bioavailable, into patients with fibrotic lung diseases such as IPF. The in vitro studies indicate that the compound GSK3008348 (herein "GSK") binds with high affinity and selectivity to the avB6 integrin, and delineate a mechanism whereby engagement of avb6 this compound promotes the internalization and lysosomal-dependent degradation of the integrin with subsequent decreases in TGF-b1 activation and Smad2 phosphorylation. The potential efficacy is supported with in vivo studies using CT/SPECT to demonstrate the competitive inhibition of a labeled avB6 ligand in the lungs by the GSK compound (both at baseline and following bleomycin-administration) and in a study demonstrating that prophylactic subcutaneous administration of the compound can diminish murine lung fibrosis in the bleomycin model. This prophylactic effect in vivo is supported by decreased levels of surrogates of fibrosis in lung tissue and BAL cells. Additional supportive data is included from patient samples (precision-cut lung slices and BAL) that show similar compound-mediated reductions in TGF-b1 activation, Smad2 phosphorylation and surrogate markers of fibrosis.

Overall, the manuscript is well written, the presented studies are well designed and executed, the data are clearly presented and accurately interpreted. Conceptually, therapeutically targeting integrin avb6 in the context of fibrotic lung disease is not inherently novel, but the introduction of an inhaled compound with efficacy and the delineation of how this compound promotes integrin internalization and lysosomal degradation are innovative and important to the field. However, some aspects of the study design could diminish the potential for translational impact and should be addressed:

1) First, the studies supporting the mechanism of integrin internalization and degradation via the lysosomal pathway in response to binding of the GSK compound (Figures 3 and 5) were done solely in normal human bronchial epithelial cells. However, epithelial cells in patients with IPF (and other fibrotic lung diseases) have impaired proteostasis that is, at least in part, due to impaired autophagy. Because autophagy and lysosomal mediated protein degradation pathways can interact it is possible the effects of this compound will be diminished in primary cells from patients with the disease of interest. It is, therefore, important to demonstrate that the compound maintains activity in cells from patients with the disease and to determine whether the efficacy requires intact autophagy and proteostatic cellular machinery.

2) The potential impact is also limited by the sole use of a prophylactic murine model of lung fibrosis and the lack of data to support anti-fibrotic efficacy of the drug when given via inhalation as indicated in the title of the manuscript. The authors discuss that the use of this model was intended to assess target engagement and effects on TGF β 1 signaling, and in that regard the data are supportive. However, the study is predicated on the novelty of inhalational delivery of the compound, and pre-clinical data demonstrating efficacy as an anti-fibrotic are warranted. The data provided do not support the “comprehensive assessment of both the anti-fibrotic potential and target engagement...” as a strength of the study as written in the discussion. The authors also note that phase 1 studies with normal human volunteers have been published and cite a prior study showing that topical delivery of drugs via inhalation is feasible in patients with IPF. Nevertheless, proof of principle for efficacy of inhalational administration in a disease model would substantially increase the impact of this paper. Interestingly, the “materials and methods” section discusses therapeutic protocols with intranasal administration of the compound beginning at days 7, 14 or 28 after bleomycin, but these experiments are not included in this manuscript.

Minor points:

1. Figure 5A appears to be mis-labeled, as it appears to show that each of the compounds tested decreased the inhibition of Smad2 phosphorylation as the compound concentration increased; the reviewer suspects that the Y-axis should be % of maximal (or baseline) pSmad2.
2. Only male mice were used in the studies.
3. In addition to some of the bleomycin experiments, other experimental protocols are mentioned in the “materials and methods” but not in the results (platelet aggregation assays).

Reviewer #3 (Remarks to the Author):

The authors identify a novel inhibitor of integrin α v β 6 and its ability to activate latent TGF β . It is already known that integrin α v β 6 activates latent TGF β in lung. There has been in fact an antibody by Biogen in Phase II trials. The overall approach and data contained herein is therefore not novel and is an incremental step over what is already known. Also, as the authors point out other integrins activate latent TGF β . This would explain the relative lack of efficacy in patients, and why antibodies that crossreact with α v β 1 are in development (to deal with the issues of functional redundancy in patients).

The authors thus have provided a novel inhibitor, but it is unclear as to whether this is actually of clinical value. In this regard, the authors need to explain Figure 2 completely with regard to the

ability to crossreact with other RGD integrins, which would be expected to functionally replace α 5 β 1 in vivo. Are other non canonical pathways affected?

That said, the data contained within the paper are thorough, which would be expected given the incremental advance; although the writing is clear, the assays are well done, the literature is appropriately reviewed/

Reviewer #4 (Remarks to the Author):

This is a large part of the preclinical package for GSK3008348, and the studies are generally well conducted. I recommend publication, but only after the authors comment on the following. I don't think it is necessary to do new experiments, but if some of the data are available they should be shared in the supplement. The main text needs to be significantly altered to emphasize that the compounds are not selective at the concentrations that they were evaluated. This does not preclude publication, but should be acknowledged.

There is no explanation that I could find concerning the selectivity of the integrins for GSK3008348 (hereafter abbreviated 348), other than a very brief statement in the legend:

(A) Competition binding curves of GSK3008348 against the RGD integrins (mean \pm SEM; n=4).

It is critical to define how the values were obtained. The values seen depend entirely on the activation state of the integrin, whether it was obtained with purified integrins, what were the counter-receptor, etc. Ideally, proper K_i values would be obtained under similar activation levels. However, this is likely impractical. What is important to explain is how the assays were conducted, and whether the value for α 5 β 1 is in a cellular assay (where internalization would affect the observed value). The authors should not call the result "selectivity" unless the studies were conducted under strictly controlled and uniform conditions for all integrins.

The 8348 compound had an IC_{50} around 10 pM, but was used in most assays at 0.1 to 0.25 μ M – at this concentration the compound inhibits all other integrins, except α 11 β 3. This needs to be stressed in the text. Are other integrins being internalized?

I am not convinced by the discussion:

345 Although $\alpha\beta6$ integrins are considered to drive the majority of TGF β activation in IPF, there are
346 emerging data suggesting a role for $\alpha\beta5$ (40) and more recently $\alpha\beta1$ (19). However, data
347 generated in this study utilizing diseased human IPF lung tissue with the highly selective $\alpha\beta6$
348 molecule GSK3008348 (specifically over the other RGD integrins $\alpha\beta1$, $\alpha\beta3$, $\alpha\beta5$ and $\alpha\beta8$
349 capable of activating TGF β) suggests that $\alpha\beta6$ integrins are the key integrin driving TGF β
350 activation even in end stage fibrotic lung. However, it is not possible to completely exclude the

8348 is a good inhibitor of $\alpha\beta1$ ($K_i = 2$ nM, Fig. 2) and the IC_{50} shown in Fig. 5 is similar to this value.
For every experiment, it is important for the authors to provide in the text the concentration used in
each panel, the dose and the concentration in the lung for in vivo studies. They need to compare
these with the IC_{50} values for $\alpha\beta5$, $\alpha\beta8$ and $\alpha\beta1$. The reader then can easily evaluate their
statements of the dominant role of $\alpha\beta6$.

Point-by-point response to Reviewers' comments:

Reviewer #1

C1) This study by John et al. reports on the development on a novel small molecule RGD-mimetic that inhibits integrin avb6 and, in doing so, diminishes TGF-b1 activation and impacts lung fibrosis, and the data presented could lay the pre-clinical foundation for the translation of this compound, which is inhalationally bioavailable, into patients with fibrotic lung diseases such as IPF. The in vitro studies indicate that the compound GSK3008348 (herein "GSK") binds with high affinity and selectivity to the avB6 integrin, and delineate a mechanism whereby engagement of avb6 this compound promotes the internalization and lysosomal-dependent degradation of the integrin with subsequent decreases in TGF-b1 activation and Smad2 phosphorylation. The potential efficacy is supported with in vivo studies using CT/SPECT to demonstrate the competitive inhibition of a labeled avB6 ligand in the lungs by the GSK compound (both at baseline and following bleomycin-administration) and in a study demonstrating that prophylactic subcutaneous administration of the compound can diminish murine lung fibrosis in the bleomycin model. This prophylactic effect in vivo is supported by decreased levels of surrogates of fibrosis in lung tissue and BAL cells. Additional supportive data is included from patient samples (precision-cut lung slices and BAL) that show similar compound-mediated reductions in TGF-b1 activation, Smad2 phosphorylation and surrogate markers of fibrosis.

Overall, the manuscript is well written, the presented studies are well designed and executed, the data are clearly presented and accurately interpreted. Conceptually, therapeutically targeting integrin avb6 in the context of fibrotic lung disease is not inherently novel, but the introduction of an inhaled compound with efficacy and the delineation of how this compound promotes integrin internalization and lysosomal degradation are innovative and important to the field.

R1) We would like to thank the reviewer for their positive comments particularly about the innovation, execution of the studies and the importance of this manuscript to the field.

However, some aspects of the study design could diminish the potential for translational impact and should be addressed:

C2) First, the studies supporting the mechanism of integrin internalization and degradation via the lysosomal pathway in response to binding of the GSK compound (Figures 3 and 5) were done solely in normal human bronchial epithelial cells. However, epithelial cells in patients with IPF (and other fibrotic lung diseases) have impaired proteostasis that is, at least in part, due to impaired autophagy. Because autophagy and lysosomal mediated protein degradation pathways can interact it is possible the effects of this compound will be diminished in primary cells from patients with the disease of interest. It is, therefore, important to demonstrate that the compound maintains activity in cells from patients with the disease and to determine whether the efficacy requires intact autophagy and proteostatic cellular machinery.

R2) We have repeated these studies in small airway epithelial cells (SAECs) from both non-diseased individuals and patients with IPF. The results have been added to Fig 3. In Fig 3D panel (i and ii) we demonstrate by flow cytometric analysis that both non-diseased and IPF

SAECs show internalization of the $\alpha\beta6$ integrin at 1hr in response to treatment with GSK3008348. The magnitude of the response, approximately 50% reduction, is similar to that reported in the NHBE cells in Fig 3A(i). We accept that we have not addressed the role of autophagy in this process, however there is not a substantial difference between the internalisation of $\alpha\beta6$ between primary disease and non-disease epithelial cells, and therefore we do not believe further dissection of this mechanism would alter the primary message of this manuscript.

Indeed we recognise the importance of autophagy in fibrogenesis and believe rather than dilute the message that autophagy is important by including it in this manuscript which focuses on integrin-mediated TGF β activation, it would benefit from a unique piece of work in its own right.

C3) The potential impact is also limited by the sole use of a prophylactic murine model of lung fibrosis and the lack of data to support anti-fibrotic efficacy of the drug when given via inhalation as indicated in the title of the manuscript. The authors discuss that the use of this model was intended to assess target engagement and effects on TGF β 1 signaling, and in that regard the data are supportive. However, the study is predicated on the novelty of inhalational delivery of the compound, and pre-clinical data demonstrating efficacy as an anti-fibrotic are warranted. The data provided do not support the “comprehensive assessment of both the anti-fibrotic potential and target engagement...” as a strength of the study as written in the discussion. The authors also note that phase 1 studies with normal human volunteers have been published and cite a prior study showing that topical delivery of drugs via inhalation is feasible in patients with IPF. Nevertheless, proof of principle for efficacy of inhalational administration in a disease model would substantially increase the impact of this paper.

R3) We agree with the reviewer and have now completed these studies and included the data (Fig 7) confirming that intranasal therapeutic dosing with GSK3008348 for 14 days inhibits development of fibrosis and reduces collagen deposition as assessed by lung hydroxyproline levels and histology (H&E and Sirius red staining). The methods (p24 L19, p25 L19 and L 23 ,p27 L1), results (p13 L4) and discussion (page 17 L2) have all been updated to reflect the inclusion of this data.

C4) Interestingly, the “materials and methods” section discusses therapeutic protocols with intranasal administration of the compound beginning at days 7, 14 or 28 after bleomycin, but these experiments are not included in this manuscript.

R4) We apologise for this confusion. As stated in the methods these studies refer to dosing of single inhaled doses of GSK3008348 in mice which have received bleomycin 7, 14 or 28 days previously. The results are described in the Results section (p9 onwards) and in Figure 4 and 5C & D. However, to avoid similarly confusing the readers we have further clarified the timing in the results, methods and figure legends and now refer to these studies as “single dosing or single dose studies”. Reference to therapeutic protocols is now restricted to the data shown in figure 7 and relates only to the dosing of animals continuously from d14–day28 post-bleomycin challenge.

Minor points:

C5). Figure 5A appears to be mis-labeled, as it appears to show that each of the compounds

tested decreased the inhibition of Smad2 phosphorylation as the compound concentration increased; the reviewer suspects that the Y-axis should be % of maximal (or baseline) pSmad2.

R5) The reviewer is correct and we have modified the graph accordingly.

C6) Only male mice were used in the studies.

R6) We understand the emerging rationale for using both male and female mice, however for consistency with prior studies as well as practical and cost concerns only male mice were used in these studies.

C7) In addition to some of the bleomycin experiments, other experimental protocols are mentioned in the “materials and methods” but not in the results (platelet aggregation assays).

R7) Platelet aggregation assays were used to determine the IC₅₀ for GSK3008348 binding to α IIb β 3. This was mistakenly represented as a K_i that has now been corrected and highlighted in Figure 2B as well as the figure legend (p42).

Reviewer #3

C8) The authors identify a novel inhibitor of integrin α v β 6 and its ability of activate latent TGF β . It is already known that integrin α v β 6 activates latent TGF β in lung. There has been in fact an antibody by Biogen in Phase II trials. The overall approach and data contained herein is therefore not novel and is an incremental step over what is already known.

R8) The key novelty demonstrated in this manuscript is two-fold. Firstly, the development of an inhaled small molecule α v β 6 inhibitor that can be delivered at high target-specific concentrations directly to site of action in the lungs of IPF patients, which is then rapidly cleared to prevent systemic exposure. Secondly, a first in class integrin inhibitor that achieves target inhibition by internalising and degrading the α v β 6 integrin, thus inhibiting TGF β over the course of dosing.

However, the reviewer also raises an important point on the targeting of α v β 6 that is worth considering. Firstly, we have been aware for some time that Biogen were progressing an antibody (BG00011) to inhibit the α v β 6 integrin as a therapy for IPF. However, there are no available data to suggest whether strategies to inhibit α v β 6 integrins have been effective because the data on these studies are not available. However, since we received these reviews, we are aware that the Biogen antibody programme has been stopped due to toxicity associated with BG00011. It is not yet clear whether the toxicity was an on- or off-target affect but either way the toxicity found is likely to be related to the weekly administration of a subcutaneous antibody. Of central importance to our findings is our translational pharmacology package allows differentiation of an inhaled small molecule inhibitor like GSK3008348 from an α v β 6 blocking antibody such as BG00011 in a number of ways and adds significant value to this field of drug discovery from a dose selection perspective.

The high affinity (<nM) and long half-life associated with a subcutaneously delivered antibody is likely to result in high levels of α v β 6 integrin engagement, both in the target compartment and systemically, for prolonged periods during once weekly dosing of BG00011 (Horan et al., Am. J. Respir. Crit. Care Med., 2008, 177:56-65; Raghu et al., Am. J. Respir. Crit. Care Med., 2018, 197:A7785). Therefore, if the toxic effect observed with BG00011 is on-target in the

lung, it is likely due to the high and continuous engagement of $\alpha\beta6$ not being well-tolerated. The published data on BG00011 suggests the dose of 1mg/Kg inhibits $\alpha\beta6$ -mediated pSmad2 production by 70% in IPF patients and based on our data in Fig 6C that would reduce lung TGFB activation in IPF patients to levels below those detected in normal healthy lungs (Raghu et al., Am. J. Respir. Crit. Care Med., 2018, 197:A7785).

We have previously demonstrated that alveolar epithelial TGF β activation plays a key role in protective homeostatic mechanisms in the lung and prolonged high level inhibition may lead to exacerbations in this patient population due to increased inflammation (John et al Science Signaling 2016 Oct 25;9(451):ra104).

This raises an additional pertinent point in terms of the novelty of our work. There is currently no data published that shows the window for pSmad2/TGF β activation between healthy and IPF lungs in order to direct drug discovery initiatives with respect to the level of inhibition that should be targeted from a safety perspective. There remain a large number of commercial and academic investigators pursuing $\alpha\beta6$ integrins as a therapy and therefore work that helps to define an appropriate therapeutic index are crucial to advance these potential drugs safely.

C9) Also, as the authors point out other integrins activate latent TGF β . This would explain the relative lack of efficacy in patients, and why antibodies that cross-react with $\alpha\beta1$ are in development (to deal with the issues of functional redundancy in patients).

R9) There have not been any data supporting the role for redundancy in a human setting and the additional human IPF PCLS data we have generated in our revised manuscript (Figure 5E and F) suggests this redundancy is minimal even if it does exist. Our data suggest that the $\alpha\beta6$ integrin is the key integrin driving TGF β activation in IPF.

C10) The authors thus have provided a novel inhibitor, but it is unclear as to whether this is actually of clinical value. In this regard, the authors need to explain Figure 2 completely with regard to the ability to cross react with other RGD integrins, which would be expected to functionally replace $\alpha\beta6$ in vivo. Are other non-canonical pathways affected?

R10) We thank the reviewer for highlighting this important point. GSK3008348 is highly selective for the $\alpha\beta6$ integrin and we have emphasised this in the text in the Paragraph headed *GSK3008348 $\alpha\beta6$ affinity and RGD integrin selectivity p7*, by more completely describing Figure 2 as requested. In addition, we have displayed the fold-selectivity difference for each RGD integrin in Figure 2B we have added the following sentence

'For the TGF β -activating RGD integrins ($\alpha\beta1$, $\alpha\beta3$, $\alpha\beta5$ and $\alpha\beta8$), a minimum selectivity of 182-fold ($\alpha\beta8$) and maximum selectivity of 3,375-fold ($\alpha\beta3$) was demonstrated (Fig. 2B). Of particular note the inhibition constant for binding to the $\alpha\beta6$ integrin was 190-fold higher than for the $\alpha\beta1$ integrin and over 1,000-fold higher than for the $\alpha\beta3$ or $\alpha\beta5$ integrins (Fig. 2B).'

To further address this point we have included new data demonstrating that inhibition of the $\alpha\beta1$ integrin (using compound 8) and the $\alpha\beta3/5$ integrins (SB-267268) does not inhibit TGF β signalling from PCLS obtained from patients with IPF (new Figure 5F) whereas inhibition of $\alpha\beta6$ integrin is sufficient to almost completely inhibit TGF β signalling in PCLS from patients with IPF (new Figure 5E and 5F) as does the ALK5 inhibitor SB-525334. In summary, we believe these data provide compelling and novel evidence that inhibiting $\alpha\beta6$ in the IPF lung is sufficient to prevent disease driven TGF β activation and coupled with rapid lung and systemic

clearance will thereby minimise any unwanted systemic effects associated with inhibition of the other RGD integrins.

C11) That said, the data contained within the paper are thorough, which would be expected given the incremental advance; although the writing is clear, the assays are well done, the literature is appropriately reviewed.

R11) We are very grateful that the reviewer considers that our data and their reporting as robust.

Reviewer #4

C12) This is a large part of the preclinical package for GSK3008348, and the studies are generally well conducted. I recommend publication, but only after the authors comment on the following. I don't think it is necessary to do new experiments, but if some of the data are available they should be shared in the supplement.

R12) Once again we are grateful to the reviewer for their positive comments please see our specific responses below.

C13) The main text needs to be significantly altered to emphasize that the compounds are not selective at the concentrations that they were evaluated. This does not preclude publication, but should be acknowledged.

R13) The reviewer is correct to point out that as the concentration of GSK3008348 increases the selectivity for the other RGD integrins is reduced. It is worth highlighting that selectivity has been determined in binding assays to provide the most accurate measure of affinity and selectivity (see also Response 10 below). We have also defined the drop-off from K_i to IC_{50} values in a number of assays e.g. ~100-fold decrease from binding K_i to pSmad2 IC_{50} in NHBE cells, which would be expected when moving from measuring binding affinity at a protein level through to downstream signalling events in primary cells. It would also be predicted that this drop-off from binding to functional assays would be comparable between integrins. Therefore, a binding K_i of 2.6 nM for $\alpha v\beta 1$ would equate to an IC_{50} in a comparable functional system of ~260 nM. Therefore, when applying selectivity differences in functional assays for GSK3008348 this needs to be taken into account.

This is only relevant in systems where potential contributions to the effect measured can be made from non- $\alpha v\beta 6$ RGD integrins. Therefore, in the NHBE system where only $\alpha v\beta 6$ -specific end points are measured e.g. internalisation and total expression of $\alpha v\beta 6$, the effect on other integrins is not relevant and maximal effect concentrations have been selected based on full internalisation concentration-response curves determined in those systems, then applied in single concentration experiments. However, where there is the potential for other RGD integrins to activate TGF β in a test system e.g. $\alpha v\beta 1$, $\alpha v\beta 3$, $\alpha v\beta 5$ and $\alpha v\beta 8$, other controls are required to show that the effect of GSK3008348 is being mediated through $\alpha v\beta 6$. This is especially important when TGF β activation is measured in the NHBE cells and IPF human PCLS assay. These have been controlled by showing in the NHBE cell system that a truly selective $\alpha v\beta 6$ ligand (A20FMDV2) inhibits all TGF β activation compared with an ALK5 inhibitor (SB-525334) and GSK3008348. Therefore, this shows TGF β activation in this test system is mediated through $\alpha v\beta 6$ alone. For the new data (Fig 5E) obtained using the IPF human PCLS assay system we chose control compounds that inhibit the RGD integrins that have been hypothesised to be play a role in TGF β activation in IPF e.g. $\alpha v\beta 1$ and $\alpha v\beta 5$ (no selective

controls for $\alpha\beta 8$ exist). The observation that these are inactive coupled with an IC_{50} for GSK3008348 in this system that is comparable to that in the NHBE pSmad2 assay shows GSK3008348 is inhibiting TGF β activation via $\alpha\beta 6$ in the IPF PCLS assay. However, we believe this is a fair point that warrants more specific highlighting in the revised manuscript and we have emphasised this in changes in the results and discussion (p10 para 2, p11, para 2 and p18, para 2).

C14) There is no explanation that I could find concerning the selectivity of the integrins for GSK3008348 (hereafter abbreviated 348), other than a very brief statement in the legend: (A) Competition binding curves of GSK3008348 against the RGD integrins (mean \pm SEM; n=4).

It is critical to define how the values were obtained. The values seen depend entirely on the activation state of the integrin, whether it was obtained with purified integrins, what were the counter-receptor, etc. Ideally, proper K_i values would be obtained under similar activation levels. However, this is likely impractical. What is important to explain is how the assays were conducted, and whether the value for $\alpha\beta 6$ is in a cellular assay (where internalization would affect the observed value). The authors should not call the result “selectivity” unless the studies were conducted under strictly controlled and uniform conditions for all integrins.

R14) This is a very important point that the authors have specifically investigated in the past and where the pitfalls have been demonstrated (Rowedder et al., 2017, SLAS Discovery, 1-12; Hall and Slack, 2019, Biomed. Pharmacother., 110, 362-370). All radioligand binding methods applied and referenced use soluble integrin protein with a standardised 2 mM Mg^{2+} as divalent metal cation and this has been shown to induce comparable activation of the $\alpha\beta$ integrins. The following sentence has been adapted to highlight this (page 21, L18) :-

‘Radioligand binding (soluble integrin proteins and human lung parenchyma membranes) and platelet aggregation ($\alpha IIb\beta 3$ only) assays completed against the RGD-integrins were completed as previously described (34,44), with binding assays completed in the presence of 2 mM Mg^{2+} to standardise integrin activation state.’

C15) The 8348 compound had an IC_{50} around 10 pM, but was used in most assays at 0.1 to 0.25 microM – at this concentration the compound inhibits all other integrins, except $\alpha IIb\beta 3$. This needs to be stressed in the text.

R15) Please refer to Response 10.

C16) Are other integrins being internalized?

R16) There is limited evidence in literature studies for internalisation of the other RGD integrins (beyond $\alpha\beta 6$) that is as a direct result of RGD ligand engagement. There are some data but in conflicting reports for $\alpha\beta 3$ and the literature is restricted to this. We have investigated cell systems that express the $\alpha\beta$ integrins and surface expression has been measured (performed via flow cytometry similar to $\alpha\beta 6$). We observed no loss of surface expression in the presence of a range of pan- $\alpha\beta$ and selective RGD ligands suggesting RGD binding induced-internalisation is specific to $\alpha\beta 6$ (data not shown). This is perhaps not a surprise as it is $\alpha\beta 6$ that is used for a number of viruses for entering host cells via RGD binding e.g. foot-and-mouth virus.

C17) I am not convinced by the discussion:

'Although $\alpha\beta6$ integrins are considered to drive the majority of TGF β activation in IPF, there are emerging data suggesting a role for $\alpha\beta5$ (40) and more recently $\alpha\beta1$ (19). However, data generated in this study utilizing diseased human IPF lung tissue with the highly selective $\alpha\beta6$ molecule GSK3008348 (specifically over the other RGD integrins $\alpha\beta1$, $\alpha\beta3$, $\alpha\beta5$ and $\alpha\beta8$ capable of activating TGF β) suggests that $\alpha\beta6$ integrins are the key integrin driving TGF β activation even in end stage fibrotic lung.

However, it is not possible to completely exclude the ...'

8348 is a good inhibitor of $\alpha\beta1$ ($K_i = 2$ nM, Fig. 2) and the IC₅₀ shown in Fig. 5 is similar to this value. For every experiment, it is important for the authors to provide in the text the concentration used in each panel, the dose and the concentration in the lung for in vivo studies. They need to compare these with the IC₅₀ values for $\alpha\beta5$, $\alpha\beta8$ and $\alpha\beta1$. The reader then can easily evaluate their statements of the dominant role of $\alpha\beta6$.

R17) This is a fair point and we have subsequently completed further studies with selective tools to show that the response in the IPF PCLSs is driven by $\alpha\beta6$ (please also see Response 9 for further detail). For the *in vivo* section more detail on the PK of GSK3008348 is included in the PD sections to highlight that undetectable, and therefore extremely low, levels of drug are present in the lung when PD changes are observed (p10 L1). This confirms the effect is not driven by lung levels of drug but by the internalisation and degradation of target initiated by GSK3008348.

REVIEWERS' COMMENTS:

Reviewer #1 (Remarks to the Author):

The authors have addressed the fundamental concerns raised in my initial review. As a minor point, the data referenced regarding NHBE cells in figure 3D (lines 168-169 in the manuscript) are not shown in the new figure, which only shows the SAEC data.

Reviewer #3 (Remarks to the Author):

This paper continues to explore a concept initially proposed by Dean Sheppard and co-workers. The nature of this project is to explore the ability of integrins to activate latent TGFbeta. This concept is interesting in principle, as it attempts to circumvent the problems around targeting TGFbeta signaling in general, which are too numerous to enumerate here. The main issue with this project is that integrin α v β 6 affects activation in EPITHELIA. Received wisdom has understood that FIBROBLASTS are the key effector cells of fibrosis, and, in the case of fibroblasts, activation required a different integrin α v β 1. This issue has possibly led to the discontinuation of the Biogen program in α v β 6 and has led to the pursuit, by the thought leaders in the area, of antibodies that block both integrins. This project is being pursued by a Bay Area startup. The data that the authors have presented are extremely exhaustive, as would befit the resources of GSK.

of I do not find their arguments regarding the advantages of their inhaled compound over the Biogen and Sheppard information convincing. For example Figure 6A or 7C does not show a complete reversal of the bleomycin-phenotype where collagen (HYP or H and E/Sirius red) is examined. I see no data on fibroblast activation eg SMA expression. Of course, these are the effector cells of fibrosis. As I am sure the authors are aware, phospho-Smad2 is indeed a feature of TGFbeta activation, but it is noncanonical signaling downstream of integrins, notably FAK and downstream effectors, that are essential for myofibroblast activation and fibrosis.

In short, I see absolutely no evidence that fibroblasts are affected.

Reviewer #5 (Remarks to the Author):

This is a very comprehensive and well-conducted study detailing the pharmacology of a potentially important novel treatment for pulmonary fibrosis, that operates through inhibition of avb6 integrin, leading to a reduction in processing and activation of TGFb.

I have specifically reviewed the details of the responses to reviewer 4, as requested. In my opinion the authors have provided adequate response to these comments, and the reviewer had indicated that no further experimentation was necessary. I agree with this comment.

I would only add one further minor request, to improve clarity. The Table shown in Fig. 2B is a critical and central set of data for this paper, and the pharmacology of this novel inhibitor GSK3008348. However, I personally find the labeling at the top of the table to be confusing. I understand it now, but it has taken a little while. I think that the main confusion is denoting 'pKi/pIC50', which suggested initially to me that the figures in the Table may be ratios of the binding constant Ki to the IC50, which is not the case of course. It would be better to say 'pKi or pIC50*' and '(Ki or IC50* nM)'. This would help with the clarity of this important Table.

I am also still not clear why binding was not measured for allbb3. Why was this?

Other than this minor comment and suggested amendment, I am happy that the responses to reviewer 4 have been addressed satisfactorily.

RESPONSE TO REVIEWERS

Reviewer #1 (Remarks to the Author):

The authors have addressed the fundamental concerns raised in my initial review. As a minor point, the data referenced regarding NHBE cells in figure 3D (lines 168-169 in the manuscript) are not shown in the new figure, which only shows the SAEC data.

RR1 Text has been updated and reference to NHBE cells in Fig 3D removed.

Reviewer #3 (Remarks to the Author):

This paper continues to explore a concept initially proposed by Dean Sheppard and co-workers. The nature of this project is to explore the ability of integrins to activate latent TGFbeta. This concept is interesting in principle, as it attempts to circumvent the problems around targeting TGFbeta signaling in general, which are too numerous to enumerate here. The main issue with this project is that integrinalphavebeta6 affects activation in EPITHELIA. Received wisdom has understood that FIBROBLASTS are the key effector cells of fibrosis, and, in the case of fibroblasts, activation required a different integrin alphavbeta1. This issue has possibly led to the discontinuation of the Biogen program in alphaVbeta6 and has led to the pursuit, by the thought leaders in the area, of antibodies that block both integrins. This project is being pursued by a Bay Area startup. The data that the authors have presented are extremely exhaustive, as would befit the resources of GSK.

I do not find their arguments regarding the advantages of their inhaled compound over the Biogen and Sheppard information convincing. For example Figure 6A or 7C does not show a complete reversal of the bleomycin-phenotype where collagen(HYP or H and E/Sirius red) is examined. I see no data on fibroblast activation eg SMA expression. Of course, these are the effector cells of fibrosis. As I am sure the authors are aware, phospho-Smad2 is indeed a feature of TGFbeta activation, but it is noncanonical signaling downstream of integrins, notably FAK and downstream effectors, that are essential for myofibroblast activation and fibrosis. In short, I see absolutely no evidence that fibroblasts are affected.

RR2 Our data suggest that in the systems we tested epithelial derived TGFβ is sufficient to phosphorylate Smad2 in lung tissue slices that include fibroblasts, as all Smad2 phosphorylation signals could be inhibited by GSK3008348. However, we accept that we have not specifically addressed the contribution of myofibroblasts to the development of pulmonary fibrosis in this manuscript and it is therefore possible that non-canonical signals that are TGFβ independent might be important in the development of pulmonary fibrosis and would therefore not be inhibited by GSK3008348. We have therefore added the following paragraph to the manuscript. See Discussion p 18 paragraph2:

'However, this does not discount a potential role of other integrins in the pathogenesis of pulmonary fibrosis. Although epithelial cell injury may be a driver of pulmonary fibrosis, myofibroblast are a key effector cell and fibroblast to myofibroblast differentiation is a major step in this process. Active TGFβ is well known to induce myofibroblast differentiation which is mediated in part through Smad

signalling, but also by non-canonical activation of Focal Adhesion Kinase (FAK) through interactions with integrins (Thannikal et al <https://www.jbc.org/content/278/14/12384>). Therefore, even if epithelial cells do generate sufficient active TGF β to lead to downstream TGF β signalling in neighboring fibroblasts, as these data suggest, it is possible that myofibroblasts develop autonomous, non-canonical, pro-fibrotic pathways through TGF β induced α v β 1 integrin and FAK activation that will not be inhibited by GSK3008348. Therefore, further studies are required to determine the relative contribution of α v β 1 and α v β 6 integrins on myofibroblast differentiation.'

Reviewer #5 (Remarks to the Author):

This is a very comprehensive and well-conducted study detailing the pharmacology of a potentially important novel treatment for pulmonary fibrosis, that operates through inhibition of α v β 6 integrin, leading to a reduction in processing and activation of TGF β . I have specifically reviewed the details of the responses to reviewer 4, as requested. In my opinion the authors have provided adequate response to these comments, and the reviewer had indicated that no further experimentation was necessary. I agree with this comment. I would only add one further minor request, to improve clarity. The Table shown in Fig. 2B is a critical and central set of data for this paper, and the pharmacology of this novel inhibitor GSK3008348. However, I personally find the labeling at the top of the table to be confusing. I understand it now, but it has taken a little while. I think that the main confusion is denoting 'pKi/pIC50', which suggested initially to me that the figures in the Table may be ratios of the binding constant K_i to the IC50, which is not the case of course. It would be better to say 'pKi or pIC50*' and '(K_i or IC50* nM)'. This would help with the clarity of this important Table. I am also still not clear why binding was not measured for allbb3. Why was this?

Other than this minor comment and suggested amendment, I am happy that the responses to reviewer 4 have been addressed satisfactorily.

RR3 The table headings have been modified to improve clarity as requested. In addition, in response to the additional question from Reviewer 5 regarding the lack of binding assay data for allbb3, no radioligand small molecule that bound with sufficiently high affinity to the allbb3 integrin was available. Therefore a platelet aggregation assay was used to show lack of activity of GSK3008348 for this integrin instead.